# Expanding the Knowledge of the Molecular Effects and Therapeutic Potential of Incomptine A for the Treatment of Non-Hodgkin Lymphoma: In Vivo and Bioinformatics Studies, Part III

**DOI:** 10.3390/ph18091263

**Published:** 2025-08-25

**Authors:** Normand García-Hernández, Fernando Calzada, Elihú Bautista, José Manuel Sánchez-López, Miguel Valdes, Claudia Velázquez, Elizabeth Barbosa

**Affiliations:** 1Unidad de Investigación Médica en Genética Humana, UMAE Hospital Pediatría 2° Piso, Centro Médico Nacional Siglo XXI, Instituto Mexicano del Seguro Social, Av. Cuauhtémoc 330, Col. Doctores, Mexico City 06725, Mexico; 2Unidad de Investigación Médica en Farmacología, UMAE Hospital de Especialidades, 2° Piso CORSE, Centro Médico Nacional Siglo XXI, Instituto Mexicano del Seguro Social, Av. Cuauhtémoc 330, Col. Doctores, Mexico City 06725, Mexico; valdesguevaramiguel@gmail.com; 3CONAHCYT-División de Biología Molecular, Instituto Potosino de Investigación Científica y Tecnológica A. C, San Luis Potosí 78216, Mexico; francisco.bautista@ipicyt.edu.mx; 4Hospital Infantil de Tlaxcala, Investigación y Enseñanza, 20 de Noviembre S/M, San Matias Tepetomatitlan, Apetatitlan de de Antonio Carvajal, Tlaxcala 90606, Mexico; cienciaflosan@gmail.com; 5Phagocytes Architecture and Dynamics, IPBS, UMR5089 CNRS-Université Toulouse 3, 205 Route de Narbonne, 31077 Toulouse, France; 6Instituto Politécnico Nacional, Sección de Estudios de Posgrado e Investigación, Escuela Superior de Medicina, Plan de San Luis y Salvador Díaz Mirón S/N, Col. Casco de Santo Tomás, Miguel Hidalgo, Mexico City 11340, Mexico; rebc78@yahoo.com.mx; 7Laboratorio de Inmunología, Departamento de Sistemas Biológicos, Universidad Autónoma Metropolitana, Unidad Xochimilco, Calz. del Hueso 1100, Col. Villa Quietud, Coyoacán, Mexico City 04960, Mexico; 8Área Académica de Farmacia, Instituto de Ciencias de la Salud, Universidad Autónoma del Estado de Hidalgo, Circuito exHacienda La Concepción S/N, Carretera Pachuca-Atocpan, San Agustin Tlaxcala 42076, Mexico; cvg09@yahoo.com

**Keywords:** incomptine A, non-Hodkin lymphoma, antilymphoma activity, Tandem Mass Tag, network pharmacology, molecular docking, toxicoinformatic, pharmaceutical analysis

## Abstract

**Background/Objectives**: Non-Hodgkin lymphoma (NHL) is a group of blood cancers that arise in the lymphatic nodes and other tissues after an injury to the DNA of B/T lineage and NK lymphocytes. Recently, we reported that incomptine A (**IA**) has in vivo antilymphoma properties. This research aimed to evaluate the effects of **IA** in the treatment of NHL using antilymphoma activity, Tandem Mass Tag (TMT), and bioinformatics approaches. **Methods**: The antilymphoma activity of **IA** was tested on male Balb/c mice inoculated with U-937 cells. Also, TMT, gene ontology enrichment, Reactome pathway, Kyoto Encyclopedia of Gene and Genomes pathway, molecular docking, toxicoinformatic, and pharmaceutical analyses were performed. **Results**: By TMT analysis of the altered levels of proteins present in the lymph nodes of Balb/c mice with NHL and treated with **IA**, we identified 106 significantly differentially expressed proteins (DEPs), including Il1rap, Ifi44, Timd4, Apoa4, and Fabp3 as well as Myh3, Eno 2, and H4c11. Among these, the Fhl1 result was the most important cluster altered and a potential core target of **IA** for the treatment of NHL. Network pharmacology studies have revealed that DEPs are associated with processes such as muscle contraction, glycolysis, hemostasis, epigenetic regulation of gene expression, transport of small molecules, neutrophil extracellular trap formation, adrenergic signaling in cardiomyocytes, systemic lupus erythematosus, alcoholism, and platelet activation, signaling, and aggregation. Computational studies revealed strong binding affinities with six proteins associated with cancer, positive pharmacokinetic properties, and no toxicity. **Conclusions**: Our contribution suggests that **IA** may be a compound with potential therapeutic effects against NHL.

## 1. Introduction

Non-Hodgkin lymphoma (NHL) is a kind of blood cancer that generally develops in lymph nodes or in lymphatic tissue present in the organs of the human body such as the skin, intestines, or stomach. NHL is characterized by the malignant transformation of NK, T, or B cells and is associated with an increased proliferation and reduction in apoptosis [1]. NHL affects people of all ages, but is rare in children. There were 544,352 new cases and 259,793 deaths around the world in 2020; among these, 112,576 were women and 147,217 were men [1,2]. In Mexico, it was reported that NHL was the ninth leading cause of cancer for both sexes and was the fourth cause of death among neoplasms of lymphatic tissue and blood-forming and related organs [3]. Treatment is according to the subtype and stage of disease; usually, chemotherapy, stem cell transplantation, immunotherapy, and radiation therapy are the principal ways to treat NHL. In relation to chemotherapy, the drugs currently employed are effective but expensive, and they show several side effects such as mouth sores, nausea, vomiting, diarrhea, constipation, bladder irritation, and blood in the urine [4,5]. Methotrexate, an antimetabolite compound, is the first-line drug to treat NHL in Mexico; however, it causes hepatotoxicity, nephrotoxicity, dermatologic toxicity, and genotoxic damage, among others [5]. Therefore, it is important to understand the processes involved in the development of NHL and search for novel drugs that are easy to obtain, cheap, with low side effects, more effective, and that have therapeutic potential to treat several neoplasms. In this sense, recently the use of proteomic, network pharmacology, and computational studies has attracted increasing attention for finding novel drugs. Also, the integration of studies that include in vivo and in vitro activity associated with proteomic studies, network pharmacology, and in silico approaches have provided an important research strategy to know and elucidate the mode of action, find new targets, discover new biomarkers of the disease, and determine pharmacokinetic properties and toxicity, among others [6,7,8,9,10,11,12,13,14,15]. 

Incomptine A (IA), a heliangolide sesquiterpene lactone isolated from the roots and leaves of *Decachaeta incompta* (DC) R. M. King and H. Robinson [16], has been reported with several biological properties such as antiprotozoal, antibacterial, antipropulsive, anti-inflammatory, allelopathic, spermatic, antimicrobial, and phytotoxic effects as well as cytotoxic and antitumor activity (Figure 1 and Figure 2). In relation to its cytotoxic properties, IA exhibited dose-dependent activity against four subtypes of NHL cells (U-937, Farage, SU-DHL-2, and REC-1), three subtypes of leukemia cells (Hl-60, K562, and REH), and six subtypes of breast cancer cells (4T1, MDA-MB-231, SK-BR-3, T-47D, MCF7, and MCF10A). Also, it regulates NF-kB expression, induces apoptosis, induces the production of reactive oxygen species, and induces the differential protein expression of cytoskeleton proteins and glycolytic enzymes in U-937 cells and non-Hodgkin lymphoma in mice [10,16]. Additionally, IA could affect the antiapoptotic function of hexokinase II in 4T1 breast cancer cells [17].

As a part of our study of the antitumor properties of **IA** and search for new natural anticancer products [10,17], we reported the molecular effects and therapeutic potential of **IA** for the treatment of NHL using antilymphoma activity, Tandem Mass Tag (TMT), gene ontology enrichment, Reactome pathway, Kyoto Encyclopedia of Gene and Genomes pathway, molecular docking, and toxicoinformatic and pharmaceutical analysis.

## 2. Results

### 2.1. Isolation and Identification of Sesquiterpene Lactone Incomptine A from Decachaeta Incompta 

Dichloromethane extract from *Decachaeta incompta* (DCEDi) leaves was analyzed to detect the presence of sesquiterpene lactone **IA** by HPLC-DAD, using as standard an authentic sample obtained previously [16]. The results showed the presence of the heliangolide **IA** at 44.9 min (Figure 3) (Table 1). Incomptine A was obtained as white crystals from CH_2_Cl_2_-hexane with Rf of 0.60 and mp of 176–177 °C. **IA** was identified by comparison of the ^1^H and ^13^C NMR data (Table 2) with the spectroscopic data obtained from the authentic sample. In the ^13^C-NMR spectra, 17 signals were observed. The ^1^H and ^13^C NMR (Table 2) showed the presence of the *α*, *β*-unsaturated *γ*-lactone by the signals for an exocyclic methylene (*δ*_H_ 6.40, 1H, dd, *J* = 2.0, 0.5 Hz and 5.80, 1H, d, *J* = 2.0 Hz) and a *γ*-lactone carbonyl (*δ*_C_ 169.7, C). These spectra showed signals for a trisubstituted double bond (*δ*_H_ 5.31, 1H, dq, *J* = 11.0, 1.5 Hz, CH; *δ*_C_ 126.1, CH, *δ*_C_ 136.0, C), a disubstituted double bond (*δ*_H_ 5.55, 1H, dd, *J* = 11.5, 7.5 Hz, and *δ*_H_ 6.14, 1H, d, *J* = 11.5 Hz; CH; *δ*_C_ 128.3, *δ*_C_ 132.3; CH), an epoxy group (*δ*_H_ 3.26, 1H, dd, *J* = 7.5, 1.0 Hz, CH; *δ*_C_ 60.4, CH, *δ*_C_ 61.2, C), two methyl groups (*δ*_H_ 1.41, 3H, s; *δ*_C_ 19.7, CH and *δ*_H_ 1.88, 3H, s; *δ*_C_ 23.7, CH), and one acetyl group (*δ*_H_ 2.02, 3H, s, *δ*_C_ 20.7, CH; *δ*_C_ 169.2, C). Finally, all the signals of the ^1^H and ^13^C NMR spectra obtained were compared with those reported in the literature [16] and were confirmed by MNova from MestreLab (version 14.0). Also, a presumptive identification was made by thin-layer chromatography (Figure 4); the same retention factor was obtained comparing with an authentic sample [16].

### 2.2. In Silico Prediction of Pharmacokinetics and Toxicological and Physicochemical Properties of Incomptine A and Methotrexate 

The absorption, distribution, metabolism, excretion, and toxicity (ADMET) properties of incomptine (**IA**) and methotrexate (**MTX**) were predicted utilizing ADMETlab, SwissADME, admetSAR, and PROTOX web servers (accessed on 13 July 2025). It is important to highlight that in recent years, the bioinformatic analysis to predict the ADMET properties of compounds under drug development has increased and has become an indispensable tool in the development of new drug candidates [13,18,19]. In this context, ADMET properties were calculated for **IA** and **MTX** to compare them (Table 3). The analysis suggests that **IA** has better oral absorption than MTX. In relation to the metabolism of **IA** and **MTX**, several CYP450 isoforms were evaluated to know if the molecules were substrates and/or inhibitors of them. Results suggest that both compounds can be metabolized by CYP3A4; they did not act like inhibitors of any CYP isoforms evaluated. In respect to distribution, the results suggest that **IA** has a better volume of distribution than **MTX,** and their excretion results suggest that IA may have a higher half-life than **MTX**. Finally, results of toxicity suggest that **MTX** may be more toxic and cause hepatotoxicity, neurotoxicity, and respiratory toxicity. In addition, the median lethal dose to humans predicted that **MTX** is more lethal than **IA,** in agreement with their category classes, I and IV, respectively.

Physicochemical properties are important parameters measured in medicinal chemistry to know the probability of the molecules submitted for study to be a candidate for the development of new drugs and their possibility of being administered orally (Table 4). According to the results, **IA** proved to be a better candidate for an orally administered drug than **MTX** because it may have a better solubility than MTX (LogP and LogS); moreover, the solubility class is better for **IA** than **MTX** according to the ADMET results, which indicated better absorption for **IA**. Drug-likeness analysis showed that **IA** fulfills all the important characteristics for an orally administered drug satisfactorily in comparison with **MTX,** which violates three important filters (Veber, Egan, and Muegge). 

### 2.3. Cytotoxic Activity of Incomptine A and Methotrexate 

The cytotoxic activities of incomptine A (**IA**) and methotrexate (**MTX**) were assayed on U-937 cells (histiocytic lymphoma) by MTT test. To compare the cytotoxic activity of the sesquiterpene lactone **IA**, we included **MTX** as the positive control, considering that it is an antimetabolite drug used in Mexico for the treatment of non-Hodgkin lymphoma [5]. The results for cytotoxic activity on the U-937 cells showed that **IA** and **MTX** had dose-dependent cytotoxic effects (Figure 5). In addition, **IA** showed a better cytotoxic activity compared with **MTX** (Table 5).

### 2.4. Antilymphoma Activity of Sesquiterpene Lactone, Incomptine A, and Methotrexate

To evaluate the antilymphoma effect of sesquiterpene lactone, incomptine A (**IA**), and methotrexate (**MTX**), the weight of the lymph nodes of male mice with NHL were obtained, and the percentages of inhibition of lymph nodes after treatment were calculated. Table 6 shows the antilymphoma activity in mice treated with **IA** and **MTX.** Both compounds had strong antitumor activity, with ED_50_ values of 7.5 mg/kg and 1.4 mg/kg, respectively. **MTX** was more potent than **IA**.

In addition, the proteomics study was performed at a dose of 10 mg/kg; the antilymphoma activity was studied in these conditions. **IA** showed 62% growth inhibition of lymph nodes in male Balb/c mice. In the case of **MTX**, it was not determinate, because a dose of 10 mg/kg caused 100% mortality in the first week of the test. Therefore, for the proteomic study, for **MTX**, the percentage of growth inhibition in lymph nodes was obtained at a dose of 1.25 mg/kg; the result was 33.4% growth inhibition in male Balb/c mice and 20% mortality.

### 2.5. Identification of Differentially Expressed Proteins in Lymph Nodes from Male Balb/C Mice with Non-Hodgkin Lymphoma and Treated with Incomptine A 

To delve deeper and explore the changes induced in the proteome of the lymph nodes from male Balb/c mice with non-Hodgkin lymphoma and treated with **IA** (LNNHLTIA) or **MTX** (LNNHLTMTX), compared with lymph nodes of male Balb/c mice with non-Hodgkin lymphoma (LNNHL), the differentially expressed proteins (DEPs) were identified by a TMT-based labeling approach coupled with UHPLC-MS/MS.

The lymph nodes were clustered as pools of all mice in each group, including the 10 mg/kg and 1.25 mg/kg **IA** and **MTX** treatments, respectively. The lymph node samples were lysed, and then the protein was alkylated and digested. The peptide samples were labeled with TMT reagent and subjected to UHPLC-MS/MS; six raw MS files with 20,453 peptides were compared against the mouse protein database. Among these, a total of 2717 proteins were identified and quantified in the lymph node samples. The DEPs were quantified according to the values of fold change (FC); FC > 1.5 was considered upregulated, while FC < 0.66 was considered downregulated (Table 7). The raw proteomics data have been deposited into the ProteomeXchange Consortium via the PRIDE, through px-submission-tool (version 2.7.3) partner repository under the dataset identifier PXD060392. 

In total, we obtained 106 significant DEPs, including 66 downregulated and 40 upregulated; among these, **IA** induced downregulation in 66 proteins and upregulation in 40. Also, 42 were altered by **IA** only (Table 8), and 64 overlapped in **IA** and **MTX** (Table 9). **MTX** induced downregulation in 57 and upregulation in 7 proteins (Table 8). In addition, the 106 DEPs gained were used for the next step of research on potential targets and the effect of **IA** in the treatment of NHL.

### 2.6. Bioinformatic Analysis

#### 2.6.1. Protein–Protein Interaction Network of Incomptine A Potential Targets for the Treatment of Non-Hodgkin Lymphoma 

In order to find the most important protein clusters altered and potential core targets of incomptine A (**IA**) in the treatment of non-Hodgkin lymphoma (NHL) in male Balb/c mice, a total of 106 DEPs (Table 8 and Table 9), containing 66 downregulated and 40 upregulated proteins, were investigated by the protein–protein interaction (PPI) enrichment networks using Cytoscape as well as the String database (Figure 6). A large number of interactions were found to be enriched, including Fhl1, Il1rap, Ifi44, Timd4, Apoa4, Fabp3, Myh3, Eno 2, and H4c11. Among these, Fhl1 resulted as the most important DEP; it was shown directly or indirectly interacting with 101 other DEPs, containing other DEPs with fewer interactions such as Il1rap, Ifi44, Timd4, Apoa4, Fabp3, Myh3, Eno 2, and H4c11. Fhl1 was a downregulated protein, as were Fabp3, Ifi44, Myh3, and Eno 2. H4c11, Apoa4, Il1rap, and Timd4 were upregulated proteins. Also, four DEPs, including Prorsd1, Ccdc90b, Gid8, and Spg20 proteins, showed no connection with other proteins.

#### 2.6.2. Results of the Enrichment Study Performed with G: Profiler, Gene Ontology, Reactome, and Kyoto Encyclopedia of Genes and Genomes

To visually disclose the enrichment pathways of the 106 DEPs, g: Profiler, gene ontology (GO), Reactome (REAC), and Kyoto Encyclopedia of Genes and Genomes (KEGG) analyses were carried out for a bioinformatics analysis. A large number of pathways were enriched according to g: Profiler analysis of the GO, REAC, and KEGG databases (Figure 7A).

In total, 190 terms were significantly enriched with 179, 8, and 3 related with GO, KEGG, and REAC, respectively. In the case of GO, 25, 115, and 39 terms corresponded with molecular function (MF), biological process (BP), and cellular component (CC), respectively. The top 30 significantly enriched terms by GO, KEGG, and REAC are displayed in Figure 7B.

#### 2.6.3. Results of the Gene Ontology Enrichment Study

Gene ontology (GO) enrichment analysis was further utilized to explore the enriched biological process (BP), cellular component (CC), and molecular function (MF) terms. For the BP terms, muscle system process and muscle contraction were the most abundant among the top 10 biological processes (Figure 8). In CC terms, contractile fiber, myofibril, sarcomere, and actin cytoskeleton were the most abundant among the top 10 cellular components. Finally, the most abundant terms were actin binding and actin filament binding among the top 10 MF terms (Figure 8). 

#### 2.6.4. Gene Ontology Enrichment Analysis and Kyoto Encyclopedia of Genes and Genomes Pathway Analysis

The GO and KEGG enrichment analyses were carried out through the DAVID database to forecast the pathway and the functional annotation enrichment related to incomptine A (IA) for the treatment of NHL in male Balb/c mice. The full analysis of 106 DEPs performed through Gene ontology (GO) functional analysis yielded 179 entries, with 115 associated with biological process (BP), 39 under cellular component (CC), and 25 under molecular function (MF). These pathways showed the 10 most important pathways that are affected by the 105 DEPs. Figure 9 shows the top 10 BP, CC, MF, and KEGG terms. The primary BP included muscle system process and muscle contraction (A). In the case of CC, it included contractile fiber and myofibril (B). Primary MF comprised actin binding (C).

KEGG analysis was associated with cytoskeleton in muscle cells and motor proteins (D). Necroptosis (Figure 10A), neutrophil extracellular trap formation, cardiac muscle contraction, and dilated cardiomyopathy (Figure 10B), among others, are some pathways affected by downregulated proteins such as Pgam5, Slc25a4, and H2ax as well as Myl2, Myl3, Atp2a1, and Tpm2.

#### 2.6.5. Results of the Reactome Enrichment Analysis

The analysis of Reactome (REAC) exhibited nine important networks: glycolysis, glucose metabolism, metabolism of carbohydrates, transport of small molecules, striated muscle contraction, muscle contraction, hemostasis, platelet activation signaling and aggregation, and transcriptional regulation by small RNAs, among others, were enriched in several DEPs (Figure 11). 

Among these, glycolysis, glucose metabolism, and metabolism of carbohydrates were enriched in three downregulated genes, Eno2, Eno3, and Pgam2. Transport of small molecules was enriched in five genes; among these, Mb, Slc4a1, and Slc25a4 were downregulated, and Apod and Apoa4 were upregulated. Muscle contraction and striated muscle contraction were enriched in 14 downregulated genes: Tmod4, Tnnc2, Myh3, Myh8, Tpm1, Mybpc2, Myl3, Actn3, Myl1, Tpm2, Myl2, Acta1, Tnnt3, and Tnn12. Hemostasis was enriched in three downregulated genes containing Tgfb1, Actn2, and Atp2a1. Platelet activation signaling and aggregation was enriched in one downregulated gene, Tgfb1. Finally, transcriptional regulation by small RNAs was enriched in four upregulated proteins containing H4c11, Polr2e, H3f3b, and H2ax. 

#### 2.6.6. Molecular Docking Studies of the Sesquiterpene Lactone, Incomptine A Against Six Differentially Expressed Proteins in Non-Hodgkin Lymphoma in Mice

We carried out molecular docking studies with the aim of deepening the antilymphoma properties of the sesquiterpene lactone, incomptine A (**IA**). In this sense, we considered the previous results obtained here, containing those of the TMT, GO, KEGG, REAC and PPI analyses. Molecular docking studies were performed to obtain additional information about the interactions between sesquiterpene lactone, **IA** with six DEPs; methotrexate (**MTX**), an antilymphoma drug currently used for the treatment of NHL in Mexico, was used as a control ligand. The complete analysis of molecular docking studies suggested that **IA** presents more affinity to all DEPs selected for this study than **MTX**, including Apoa4 (−6.35 Kca/mol), Fabp3 (−7.17 Kca/mol), Fhl1 (−6.86 Kca/mol), IFi44 (−6.55 Kca/mol), Il1rap (−7.15 Kca/mol), and Timd4 (−7.24 Kca/mol), in comparison with **MTX**, Apoa4 (−4.27 Kca/mol), Fabp3 (−4.84 Kca/mol), Fhl1 (−4.92 Kca/mol), IFi44 (−5.72 Kca/mol), Il1rap (−6.02 Kca/mol), and Timd4 (−5.11 Kca/mol), respectively. Furthermore, it is important to denote that in almost all proteins, the binding site is the same for **IA** and **MTX,** and they share several aminoacidic residues with mostly polar interactions (Table 9). The only protein where **IA** did not share the same binding position with **MTX** was IFi44 (Table 10, Figure 12). 

## 3. Discussion

Non-Hodgkin lymphomas (NHLs) are the 10th most common malignant neoplasm that are an important cause of mortality and morbidity around the world. In this sense, nearly 544,352 persons are diagnosed with NHL annually, resulting in nearly 259,793 deaths in 2020 [1,20,21,22]. In the last ten years, NHLs have had a rise in mortality rate and incidence with 112, 576 cases in women and 147,217 cases in men [2]. In the case of Mexico, it has been reported that NHLs were the ninth cause of cancer for both genders and were the fourth cause of death [3,4].

The first-line treatments for NHLs are chemotherapy and chemoimmunotherapy, including drugs such as prednisone, cisplatin, fludarabine, doxorubicin, vincristine, methotrexate, cyclophosphamide, etoposide, and rituximab. However, several of these drugs show limited efficacy, and some types of aggressive lymphomas relapse and progress into refractory lymphoma [23,24,25]. They also have several side effects such as kidney injury, nephrotoxicity, hepatotoxicity, and myelosuppression, among others [3,5]. In this sense, the investigation of new antitumor drugs with the best efficacy and safety is very important around the world and remains a challenge.

Part of our ongoing research involves new anti-NHL drugs isolated from Mexican medicinal plants. Previously, we reported the antilymphoma activity in a mouse model at a dose of 5 mg/kg of the sesquiterpene lactone incomptine A (**IA**) [10]. To expand on these findings, in this work, the antilymphoma activity of the **IA** was evaluated at a dose of 10 mg/kg using in vitro, in vivo, and bioinformatics approaches, enhancing the translatability of our prior findings, and we assessed the impact of **IA** on additional aspects of NHL.

First, the dichloromethane extract of the leaves of *Decachaeta incompta* was subject to sequential chromatographic methods, including column chromatography and thin-layer chromatography (TLC) over silica gel, to afford a sesquiterpene lactone that was identified as **IA** by HPLC (Figure 3 and Table 1), TLC (Figure 4), and NMR methods (Table 2). The retention time, retention factor, and NMR data were the same as those for the authentic sample provided by Dr. Bautista [16]. 

Once the identification of **IA** was made, an in silico ADMET (absorption, distribution, metabolism, excretion, and toxicity) analysis was performed [18]. In silico techniques represent a fast and economic strategy that helps to reduce time and costs in research for the development of new drugs [26,27,28]. ADMET and physicochemical properties were calculated in order to compare the results of **IA** against **MTX,** considering that the first is a potential drug and the second is one drug used currently in Mexico for the treatment of cancers such as NHL. Incomptine A exhibits favorable drug-like properties. An ADMET evaluation was carried out; this kind of in silico study brings valuable information for new molecules, including several parameters that must been determined when a new drug is in development [17,27,28]. Considering the constant advancement and development of computer software for in silico techniques, these have been transformed into a valuable tool that allows researchers to determine these and other parameters reducing costs and time and avoiding the use of animals [28]. The complete analysis of the ADMET calculation suggests that **IA** may have better absorption when it is administered orally than **MTX**, which also is reflected in a better distribution of the drug. In respect to metabolism, this parameter was similar to **MTX**, and the excretion calculated for **IA** suggests that it may have a longer T_1/2_ than **MTX**. These results may support the important activity described in this study that **IA** is more active than **MTX** for NHL treatment. Moreover, the toxicities of both molecules were determined; in this sense, we observe that **IA** may have lower toxicity than **MTX,** with a predicted human lethal dose 50 (LD_50_) of 3 mg/kg for **IA** in comparison with 1330 mg/kg for **MTX.** This result suggests a wide therapeutic window for the use of **IA** in comparison with **MTX** and classifies **IA** in Class IV (300 < LD50 ≤ 2000 mg/kg) in comparison with **MTX,** which is collocated in Class I, which is considered fatal by ingestion (LD_50_ ≤ 5 mg/kg). **MTX** is known as one of the most widely used anticancer agents [29,30,31]. It is important to highlight that the result of toxicity obtained in this study of **IA** was in agreement with the in vivo toxicity reported previously and confirms the predicted findings by bioinformatics [32]. In contrast, it is also known that the use of **MTX** may generate side effects such as nephrotoxicity, muscle pain, red eyes, swollen gums, and hair loss, among others [33,34]. It also has been reported that high doses of **MTX** may be toxic to humans and mice [5,35]. This is according to some predictions obtained from ADMET studies and represents an advantage in the use of **IA** as a drug for the treatment of cancer. Commonly, **the** route of administration of **MTX** is intravenous, which is painful for the patients [36]; in comparison, **IA** represents one important alternative because our results suggest better oral absorption with important pharmacological anticancer activity, and perhaps with lower side effects, than **MTX**. Finally, the drug likeness analysis supports the information showed in the ADMET study, because it suggests that **IA** fulfills the necessary criteria to be an important candidate for the development of an orally administered drug. 

After the ADMET analysis of **IA** was made, its cytotoxic activity was determined using U-937 cells (histiocytic lymphoma) that are associated as a cause of NHL [37]. The results showed that **IA** was nearly six times most active than **MTX** (Table 5 and Figure 5), used as positive control. Cytotoxic activity of the sesquiterpene lactone, **IA,** was more significant than those reported for other natural products with important cytotoxic properties such as farnesol, nerol, citral, and geraniol [38]. The cytotoxic activity here reported for **IA** is in agreement with previous reports about its cytotoxic properties on several cancer cell lines, including four subtypes of NHL cells (U-937, Farage, SU-DHL-2, and REC-1), three subtypes of leukemia cells (Hl-60, K562, and REH), and six subtypes of breast cancer cells (4T1, MDA-MB-231, SK-BR-3, T-47D, MCF7, and MCF10A) [10,17]. In addition, it is important to highlight that it has been reported that **IA** was not cytotoxic to the MCF10A normal human mammary breast cell line [17]. These results of ADMET and cytotoxicity suggest that **IA** has antitumor potential [39,40,41,42,43].

Then, antilymphoma activity was tested by comparing the effect of **IA** and the antilymphoma drug **MTX**, which was chosen considering that currently it is used in Mexico to treat NHL [3]. The results showed that **IA** was less active that **MTX** (Table 6); however, the activity of **IA** was similar to other natural products with significant antilymphoma properties, including flavonoid glycosides and sesquiterpenoids [10,35,38]. In addition, **IA** is a sesquiterpene lactone, a group of compounds with an *α*-methylene-*γ*-lactone moiety in their structure that have been reported as a new promising form of cancer therapy based on their anticancer activity. The most important examples include eupatolide, deoxyelephantopin, and parthenolide. In the last case, it is undergoing cancer clinical trials [39]. It is important to mention that although **MTX** was more potent in this study, it caused mortality in the treated mice at doses ≥ 1.25 mg/kg; in particular, a dose of 10 mg/kg caused 100% mortality in the first week of the test. Also, it has been reported to have serious side effects including pancreatitis, cirrhosis, aplastic anemia, leukopenia, and gastrointestinal bleeding [43]. 

The next step of this work was to delve deeper and explore the changes induced in the proteome of the lymph nodes from male Balb/c mice with non-Hodgkin lymphoma and treated with **IA** at doses of 10 mg/kg. Recently, we reported a proteomic study in a mouse model at a dose of 5 mg/kg of incomptine A [10]. To expand on these findings, in this work, the antilymphoma activity of the **IA** was evaluated at a dose of 10 mg/kg using in vitro, in vivo, and bioinformatics approaches, enhancing the translatability of our prior findings and assessing the impact of **IA** on additional aspects of NHL.

The results of antilymphoma activity showed that IA caused 62% growth inhibition of lymph nodes in male Balb/c mice. The case of **MTX** was not determinate because the dose of 10 mg/kg caused 100% mortality. Therefore, the proteomic study for **MTX** of the lymph nodes was obtained at a dose of 1.25 mg/kg (Table 6). TMT proteomics technology was used to identify the differentially expressed proteins (DEPs) of the lymph nodes from male Balb/c mice with non-Hodgkin lymphoma and treated with **IA** (LNNHLTIA) or **MTX** (LNNHLTMTX) and from male Balb/c mice with non-Hodgkin lymphoma (LNNHL). A total of 2717 proteins were identified and quantified in the LNNHLTIA in this study, among which 106 proteins showed significant expression differences (Table 8 and Table 9), including 66 downregulated proteins and 40 upregulated proteins. In general, **IA** induced more downregulation than upregulation of proteins in the lymph nodes of mice with NHL, 62% and 38%, respectively, in agreement with our previous study. Also, the number of DEPs at a dose of 10 mg/kg was 40% higher compared with those obtained at a dose of 5 mg/kg [10]. 

All DEPs altered by **IA** treatment were subjected to PPI analysis, showing that the four and a half LIM domains protein 1 (Fhl1) was the most important core protein (Figure 6); it showed direct or indirect interaction with other proteins, including Il1rap, Ifi44, Timd4, Apoa4, and Fabp3 as well as Myh3, Eno 2, and H4c11. The last three were reported recently to be possibly associated with the appearance of NHL [10]. According to the PPI network, GO, KEGG, and REAC analyses, the DEPs including Fhl1, Il1rap, Ifi44, Timd4, Apoa4, Fabp3, Myh3, Eno 2, and H4c11 may be associated with several processes, including muscle contraction, glycolysis, hemostasis, epigenetic regulation of gene expression, transport of small molecules, neutrophil extracellular trap formation, adrenergic signaling in cardiomyocytes, systemic lupus erythematosus, alcoholism, and platelet activation, signaling, and aggregation (Figure 7, Figure 8, Figure 9, Figure 10 and Figure 11).

Four and a half LIM domains protein 1 (Fhl1) resulted in the most important cluster altered and a potential core target of **IA** for the treatment of NHL in male Balb/c mice. In PPI, it showed direct interaction relationships with 40 downregulated proteins and 2 upregulated proteins. It also had an indirect interaction relationship with 59 DEPs. Fhl1 is widely expressed in humans, particularly in skeletal and cardiac muscle cells. Fhl1 participates in the development of the skeletal muscle and myocardium as well in the regulation of gene transcription, thyroid function, blood glucose levels, myoblast differentiation, and other biological process. Abnormal expression of Fhl1 is associated with several diseases such as skeletal muscle disease, reductive myopathy, hypertrophic cardiomyopathy, and viral infections (chikungunya and cashmere). It also regulates cell proliferation, differentiation, apoptosis, adhesion, migration, transcription, and other cellular processes, and it plays an important role in cell growth. In tumors, the expression of Fhl1 is upregulated or downregulated and plays a role in promoting or inhibiting tumor development. The expression of Fhl1 is downregulated in several cancers such as lung, prostate, breast, ovarian, colon, thyroid, brain, kidney, liver, and skin (melanoma), as well as oral cancers. In contrast, the expression of Fhl1 is upregulated in cancers such as laryngeal carcinomas, gastric signet ring cell carcinoma, lung adenocarcinoma, gastric cancer, acute promyelocytic leukemia, and glioblastoma, among others. It is important to highlight that considering the significant differences observed in the expressions of Fhl1 between different tumors, Fhl1 is advised as a prognostic marker and could be a potential target for cancer therapy research including NHL [44]. 

In relation to the proteins, Il1rap, Ifi44, Timd4, Apoa4, Fabp3, Myh3, Eno 2, and H4c11 showed ≤ 24 direct or indirect interaction relationships with other DEPs. These eight DEPs have been reported to be associated with one or more cancers including pancreatic, kidney, tongue, colorectal, breast, bone, cervical, stomach, small-cell lung, non-small-cell lung, head, neck, lung, liver, and nasopharyngeal cancer, among others, as well as myeloblastic, chronic myeloid, acute myeloid, and acute lymphoblastic leukemias. It is important to note that Il1rap, Ifi44, Timd4, Fabp3, and Eno 2 have been reported as potential therapeutic targets and potential prognostic biomarkers for several cancers [45,46,47,48,49,50,51,52,53,54,55,56,57,58,59,60]. 

Molecular docking assay was performed to obtain additional information about the interaction between the sesquiterpene lactone **IA** and the six DEPs implicated in NHL, Il1rap, Apoa4, Fabp3, IFi44, Timd4, and Fhl1. This technique has gained significant importance in the development of novel treatments for various diseases. This computational technique predicts the binding affinity of ligands to receptor proteins. The assay yields free Gibbs energy (ΔG) values, which indicate the exergonic nature of the reaction. A more negative ΔG value suggest a higher probability of ligand–protein union and interaction [61,62].

The results of molecular docking showed that **IA** has better affinity to all the proteins used in this study in comparison with **MTX** because its **ΔG** values were lower compared with those obtained from **MTX**. Moreover, in almost all proteins, the binding site as well as the polar interactions between **IA** and **MTX** were the same. This kind of interaction is important because it gives stability to the binding position. Our results suggest that both molecules bind on the same position of the proteins, and perhaps, **IA** may have better activity than **MTX** for the treatment of NHL. Also, this is the first report that involves Il1rap, Apoa4, Fabp3, IFi44, Timd4, and Fhl1 with NHL.

In agreement with the GO, REAC, and KEGG pathway as well as TMT, PPI network, and molecular docking analyses, the sesquiterpene lactone **IA** showed direct or indirect interaction with 106 depositions. Among these, Fhl1 was the most important core protein. In this context, Fhl1 could be a potential therapeutic target and potential prognostic biomarker in NHL. In addition, these results support the effects of **IA** in several processes such as muscle contraction, glycolysis, hemostasis, epigenetic regulation of gene expression, transport of small molecules, neutrophil extracellular trap formation, adrenergic signaling in cardiomyocytes, systemic lupus erythematosus, alcoholism, and platelet activation, signaling, and aggregation, among others. Also, the results of ADMET and antilymphoma activity provide additional evidence that support the therapeutic antitumor potential of **IA**. In addition, the information obtained in this work indicates that all approaches used can be employed to reliably find potential therapeutic targets and potential prognostic biomarkers to several cancers, including NHL. Additional assays, including immunohistochemistry, Western blotting, ELISA, and miRNA expression, among others, are mandatory to validate the results obtained here. Also, other assays are required to explore the way **IA** promotes lymphoma cell death in NHL, such as apoptosis (i.e., caspase activity, TUNEL staining, and annexin V/PI flow cytometry) and the effect on the glycolytic pathway (i.e., expression of aldolase A, lactate dehydrogenase, and hexokinase II). Given the aforementioned, we propose that IA can be considered a promising molecule with an exceptional pharmacological profile. Its potential application in the future as a novel pharmacological treatment for NHL, or as a foundation for the development of new molecule derivates to enhance its efficacy in treating this disease, warrants continued investigation of this molecule to acquire more information from it. 

## 4. Materials and Methods

### 4.1. Isolation of Incomptine A from Dichloromethane Extract of Decachaeta Incompta 

The sesquiterpene lactone **IA** was obtained from the aerial parts of *Decachaeta incompta* (Asteraceae), collected in February 2022 in the State of Oaxaca, Mexico. The leaves of the plant were botanically identified by the MS Abigail Aguilar Contreras biology of the Instituto Mexicano del Seguro Social. A sample (voucher 15311) was deposited at the Herbarium, IMSSM. The phytochemical procedure was performed according to a protocol previously described [16]. **IA** was identified by NMR spectra and HPLC-DAD and by comparison with an authentic sample having a purity near 99%. Dimethyl sulfoxide (DMSO) Sigma Aldrich (St. Louis, MO, USA), was used to dissolve **IA** and **MTX** (Sigma Aldrich).

### 4.2. Chemicals and Instrumentation

Triethylammonium bicarbonate buffer (1.0 M, pH 8.5 ± 0.1), tris (2-carboxyethyl) phosphine hydrochloride solution (0.5 M, pH 7.0), iodoacetamide (IAA), formic acid (FA), acetonitrile (ACN), methanol were purchased from Sigma Aldrich (St. Louis, MO, USA), trypsin from bovine pancreas Promega (Madison, WI, USA), ultrapure water (Millipore, Burlington, MA, USA), TMT 6-plex Isobaric Label Reagent, and Pierce Quantitative Colorimetric Peptide Assay Thermo Scientific (Waltham, MA, USA). were used in this work. An Ultimate 3000 nano UHPLC system was coupled to a Q Exactive HF MS equipped with a Nano spray Flex Ion Source Thermo Scientific (Waltham, MA, USA). TMT-based Quantification Analytical Service was provided by Creative Proteomics (Shirley, NY, USA).

### 4.3. Cell Culture Conditions 

U-937 cells were acquired from the American Type Culture Collection (CRL-1593,2, histiocytic lymphoma). To develop the mice model, U-937 cells were cultured at 37 °C in RPMI 1640 culture medium (GIBCO Cat: 11875-093) supplemented with 5% fetal bovine serum (GIBCO Cat: 16000044), streptomycin (100 μg/mL)/penicillin (100 U/mL), and 5% CO2. In vivo tests were realized using cell cultures at a density of 2.5 × 10^6^ cells in T75 flasks (Invitrogen, Waltham, MA, USA). Cell viability was determined by the trypan blue exclusion test. Cells were resuspended in fresh medium 24 h before treatments to ensure exponential growth. 

The cytotoxic activity of incomptine A (**IA**) against U-937 cells was assessed using the colorimetric 3-(4,5-dimethyl-2-thiazolyl)-2,5-diphenyl-tetrazolium bromide (MTT) test, Sigma Aldrich (St. Louis, MO, USA). Exponentially growing cells of U-937 cell lines were seeded in 96-well plates at a density of 5.0 × 10^3^ cells per well in 100 μL and treated with five serial concentrations between 0.1 μM and 4.0 μM of **IA** or **MTX** for 24 h under 5% CO_2_ and 95% O_2_ at 37 °C. The compounds were dissolved in DMSO; the final concentration of DMSO used was 0.1% (*v*/*v*) for each sample. Cells (U-937) treated with 0.1% DMSO served as the control group. After incubation for specified times, MTT reagent (10 μL, 5 mg dissolved in 1 mL of PBS) was added to each well and incubated for 4 h. The plates were centrifuged (10 min at 350 × g), and the purple formazan crystals of metabolized yellow tetrazolium salt by viable cells were dissolved in 150 μL of DMSO. Absorbance was quantified at 570 nm using the ELISA plate reader. Results were expressed as a percentage of viability, with 100% representing control cells treated with 0.1% DMSO alone. Then, the CC_50_ was determined. This was defined as the treatment concentration at which 50% reduction in cellular proliferation was observed. This was calculated graphically using the curve-fitting algorithm of the computer software Prism 5.03 (GraphPad, La Jolla, CA, USA). Values were calculated as means ± S.E.M. from three independent experiments, each performed in triplicate. Statistical analysis of the data was performed using one-way ANOVA, as well as Dunnett’s multiple comparison tests, with a value of *p* < 0.05 to establish a significant difference between the study groups.

### 4.4. Animals

To induce the lymphoma model, healthy male Balb/c mice (25 ± 3g) were provided by the animal house of the Centro Medico Nacional Siglo XXI, from Instituto Mexicano del Seguro Social. The research had ethical authorization by the National Committee of Scientific Research from the Instituto Mexicano del Seguro Social with registration numbers: R-2018-785-111 and R-2020-3601-186. Mice were maintained in polyvinyl cages at 22 °C and light–dark periods of 12 h, with ad libitum access to food and water in agreement with Mexican Official Norma, NOM-062-ZOO-1999 [63].

#### 4.4.1. Antilymphoma Test

Briefly, healthy male Balb/c mice were divided into 9 groups with 5 mice per group: LNNHL1, LNNHL, LNNHLTIA [LNNHLTIAa, LNNHLTIAb, LNNHLTIAc, and beingLNNHLTIAd], and LNNHLTMTX [LNNHLTMTXa, LNNHLTMTXb, and LNNHLTMTX c]. For comparison, LNNHL1, designated as the healthy group, was treated with tween 80 in water (2% *v*/*v*), and LNNHL was used as negative control only, inoculated with 1 × 10^6^ U-937 cells. LNNHLTIA and LNNHLTMTX groups were injected intraperitoneally with 1 × 10^6^ U-937 cells and treated orally for 9 days with incomptine A (**IA**, 2.5, 5.0, 7.5, and 10 mg-kg^−1^) or methotrexate (**MTX**, 1.25, 2.5, and 3.75 mg-kg^−1^), respectively. Animals were maintained under observation for 30 days, recording their daily survival and weight. Mice were sacrificed, and axillary and inguinal lymph nodes were removed and weighted. Antilymphoma activity was obtained by comparing the total lymph node weight of each group against the negative control, and the percentage of inhibition and the 50% effective inhibitory doses (ED_50_) were calculated. Lymph nodes obtained of the groups LNNHLTIA, LNNHLTMTX, and LNNHL were used in the TMT test.

The ED_50_ was calculated graphically using the curve-fitting algorithm of the computer software Prism 5.03 (GraphPad, La Jolla, CA, USA). Values were calculated as means ± S.E.M. (*n* = 5). Statistical analysis of the data was performed using one-way ANOVA as well as Dunnett’s multiple comparison tests with a value of *p* < 0.05 to establish a significant difference between the study groups.

#### 4.4.2. Non-Hodgkin’s Lymphoma Protein Expression Induced Through IA or MTX

To evaluate the effects in the proteome of the non-Hodgkin lymphoma experimental model, we clustered the lymph nodes as pools of all mice in each group, concerning **IA** and **MTX** treatments of 10 mg/kg and 1.25 mg/kg, respectively. Then, tissue was lysed using a TissueLyser after being centrifuged at low temperature for 15 min at 12,000 rpm, and the protein concentration was performed with a BCA kit. Following that, 100 μg of protein was treated with TCEP (10 mM) for 1 h at 56 °C and then alkylated with IAA (20 mM) for 1h in dark at room temperature. Then, trypsin was added (1:50) and incubated at 37 °C overnight. The peptides were brought to almost dryness and re-dissolved with TEAB (100 mM). After that, each sample of peptides was treated with anhydrous acetonitrile (41 μL) and centrifuged. The supernatant was treated with TMT Reagent using the following: (C-)/TMT, **MTX**/TMT, and **IA**/TMT. Finally, the reaction was incubated for 1 h at room temperature; then, it was treated with 5% hydroxylamine (8 μL) and incubated for another 15 min. 

#### 4.4.3. Nano UHPLC-MS/MS Analyses

The labeled peptides obtained in Section 4.4.2 were subjected to analysis by UHPLC-MS/MS using an Ultimate 3000 nano UHPLC system coupled to a Q Exactive HF mass spectrometer equipped with a Nano spray Flex Ion Source (Thermo Scientific, Waltham, MA). Briefly, the sample (2 μg) was injected onto a trap column (PepMap C18, 100 Å, 100 μm × 2 cm, 5 μm) and subjected to fractionation using an analytical column (PepMap C18, 100 Å, 75 μm × 50 cm, 2 μm). A linear gradient was used: 5 to 7% buffer B for 2 min, from 7% to 20% buffer B for 80 min, from 20% to 40% buffer B for 35 min, then from 40% to 90% buffer B for 4 min. Mobile phase A: 0.1% formic acid in water; B: 0.1% formic acid in 80% acetonitrile. For labeled samples, the full scan was performed between 350 and 1650 m/z at a resolution of 200 to 120,000 Th and a gain control of 3 to 6. The MS/MS scan was operated in Top 15 mode using a resolution of 200 to 30,000 Th; automatic gain control target 1 × 10^5^; charge state exclusion: unassigned, 1, > 6; dynamic exclusion 40 s, normalized collision energy at 32%; isolation window of 1.2 Th. 

#### 4.4.4. Protein Identification

The six raw MS files were subjected to analysis and compared against mouse protein database using Maxquant (v2.6.7.0), ProteoWizard (version 3), and Proteome Discoverer 3.2 (ThermoFisher, Waltham, MA, USA). The parameters were set as follows: The protein modifications were oxidation (M, variable), carbamidomethylation (C, fixed), and TMT-6Plex. Enzyme specificity was set to trypsin, the precursor ion mass tolerance was set to 10 ppm, the maximum missed cleavages were set to 2, and MS/MS tolerance was 0.6 Da. The distribution of all the proteins identified according to the protein mass (kDa), the distribution of 20,453 peptides identified according to the length, and the distribution of 2717 of the proteins identified according to sequence coverage, were performed. Data analytical report was carried out by Analytical Service from Creative Proteomics (New York, NY, USA) with *p* < 0.05 (indicates > 95% confidence of change in protein concentration respect of the magnitude of change) selected to designate differentially expressed proteins.

#### 4.4.5. Differentially Expressed Protein Analysis

A total of 2717 proteins were identified for this work, with (C-)/TMT and **MTX**/TMT used to determine protein differential expression between treatments MTX/TMT and **IA**/TMT establishing the ratios MTX/C, **IA**/C, and **IA**/MTX. It is important to highlight that fold-change cutoffs at protein level accepted with biological meaning are defined as 1.5 times up or down. Therefore, the screening criteria for the relative quantitation of proteins in this work were as follows: fold change < 0.67 was considered downregulated, and fold-change > 1.5 was considered upregulated. Each comparison was analyzed using enrichment processes analysis and cell signaling pathways. 

### 4.5. Bioinformatic Methodology

The log ratio data matrix was used to feed the algorithms for the analysis of cellular process enrichment networks and signaling pathways through programming language R v4.2.2 and RStudio v3.1.4, applying the clusterProfiler v4.2.2, MSigDB in R (Molecular Signatures Database, accessed 24 May 2025), enrich plot, and ggplo2 packages (accessed 24 May 2025). Also, we used symbol. mouse ID data (org.Mm.eg.db, accessed 24 May 2025), KEGG (Kyoto Encyclopedia of Genes and Genomes), gene ontology (GO), biological process (BP), molecular function (MF), and cellular component (CC), and Reactome databases. The enrichGO, compareCluster, and enrichKEGG functions were applied. The results were plotted with the functions “emapplot”, “aplot_list”, “dotplot”, and “cnetplot” for visualization. Dotplot and three onthologies plots enrichment was performed with the ClusterProfiler package of the R v4.2.2 tool, following the steps proposed by the developer, using the GO databases (BP, MF, CC) and FDR < 0.05 as the value of statistical significance. Additionally, enrichment networks were performed in Cytoscape plug-in using the ClueGo (Version 2.5.10) as well as STRING plugins (confidence cutoff of 0.4) for both analyses, following the developer’s manual using KEGG and GO databases and setting a *p* < 0.05 as the statistical significance criterion. In parallel, we performed a functional enrichment analysis, also known as over-representation analysis (ORA), using the g: Profiler tool available online, for which we used the parameters preset by the program. The log ratio (log1.5) is the log of the fold-changes, i.e., log1.5 (condition 1/condition 2). Log ratios are used/plotted in graphs for the exploratory analysis of pathways and functional enrichment of cellular processes in each of the different treatments, generating a log ratio values matrix with each protein. These data are better to show because they center around zero, giving reductions a negative value and increments a positive value, and because the log ratios are symmetric around zero [64].

#### 4.5.1. Comparison of Shared Processes and Molecules

Differentially expressed proteins obtained for each treatment were analyzed with the g: Profiler tool available online. Graphs and pathway enrichment tables and figures were generated by comparing with KEGG, GO, and Reactome databases. Enrichment plot images generated in the x-axis show the number of different enriched processes according to each database, while the y-axis plots the significance value obtained in ascending order (−log10 adj-*p*-value); a dashed cutoff at the top of the graph divides the most significant enriched cellular processes, and a black circle with a number of different processes of interest are highlighted according to each database, according to the order in which they appear in a general list of enriched processes. At the bottom is the list of the representative processes marked and arranged by significance value, the first having the highest value. The list is labeled by ID (number in the general list of the process), resource (database with which the process was enriched), term ID (identification of the process according to each platform), term name (name of the enriched process according to each database), and adj-*p*-value (significance value of the process) shown with number and color scale and arranged in a descending order. Finally, to know how many molecules and cellular processes are shared between the different conditions, the modulated proteins data and their change values were used to analyze them with the DiVenn v2.0 online tool [65,66,67]. All annotations to refer proteins are based on the UNIQID gene name or UNIPROT-ID. http://geneontology.org/; https://reactome.org/; https://www.uniprot.org/; or https://www.genome.jp/kegg/pathway.html (accessed on 24 May 2025).

#### 4.5.2. In Silico Physicochemical, Pharmacokinetic, and Toxicological Properties

To determine the physicochemical, pharmacokinetic, and toxicological properties of **IA** and **MTX**, ADMETlab [68], SwissADME [19], admetSAR [69], and PROTOX [70] servers were used. These servers work using the SMILES code for each chemical structure; therefore, the **IA** structure was drawn on ChemDraw software (version 20.0.0.22), and the SMILES code was obtained (code: C=C1C(=O)O[C@@H]2/C=C(/C)CC[C@H]3O[C@@]3(C)C[C@@H](OC(C)=O)[C@H]12), on the other hand, **MTX** SMILES code was obtained from PubChem library (CID 126941, and SMILES code: CN(Cc1cnc2nc(N)nc(N)c2n1)c1ccc(C(=O)N[C@@H](CCC(=O)O)C(=O)O)cc1). Both codes were used on in silico servers to predict the properties previously described for **IA** and **MTX**. 

#### 4.5.3. Molecular Docking Studies

Blind docking experiments were carried out in order to determine the probability of the ligands MTX and IA to bind to the proteins proposed in the study. The apolipoprotein A-IV (Apoa4, ID: P06728), fatty acid-binding (Fabp3, ID: P11404), four and a half LIM domains protein 1 (Fhl1, ID: P97447), interferon-induced protein 44 (Ifi44, ID: Q8BV66), interleukin-1 receptor accessory protein (Il1rap, ID: Q61730), and T-cell immunoglobulin and mucin domain-containing protein 4 (Timd4, IDQ6U7R4) were used as targets of the study. The crystals were retrieved from the Uniprot database (https://www.uniprot.org/) (accessed on 29 May 2025). Proteins were optimized; for this, the molecules of water and ions were eliminated and only those important for catalytic activity were preserved, the polar hydrogen atoms were added, and proteins were ionized considering a basic environment (pH = 7.4). Gasteiger charges were assigned; the computed output topologies from the previous steps were used as input files for docking simulations.

For ligands optimization, the two-dimensional (2D) structure of methotrexate (CID: 126941) was retrieved from the chemical library PubChem (https://pubchem.ncbi.nlm.nih.gov/) (accessed on 29 May 2025), for incomptine-A, and the 2D structure of the proposed ligand was drawn. Both ligands were optimized and submitted to energetic and geometrical minimization using Avogadro software (Avogadro: an open-source molecular builder and visualization tool. Version 1.2.0) [71]. 

Docking analysis was carried out using Autodock 4.2 software [72]. The search parameters were as follows: A grid-base procedure was employed to generate the affinity maps delimiting a grid box of 126 × 126 × 126 Å^3^ in each space coordinate, with a grid-point spacing of 0.375 Å. The grid centers (x, y, z) for each protein search were −28.512, 7.344, 21.938 for Apoa4; 0.815, 2.902, 1.116 for Fabp3; −1.118, −0.813, −5.054 for Fhl1; 5.872, −0.066, −7.999 for Ifi44; 30.545, 10.244, −27.227 for Il1rap; and −12.49, −4.782, −4.185 for Timd4. Lamarckian genetic algorithm was employed as scoring function with a randomized initial population of 100 individuals and a maximum number of energy evaluations of 1 × 10^7^ cycles. The analysis of the interactions in the protein–ligand complex was visualized with Drug Discovery Version 20.1.0.19295 (https://www.3ds.com/products/biovia/discovery-studio (accessed on 29 May 2025)) [73].

## 5. Conclusions

In conclusion, we used GO, REAC, and KEGG pathways as well as TMT, PPI network, ADMET, and molecular docking analyses and identified 106 differentially expressed lymph node proteins in NHL. We likewise found that Fhl1 was the most important core protein expressed in this study, suggesting that it plays an important role in the pathology of NHL. Also, the results suggest that Fhl1 could be a potential therapeutic target and potential prognostic biomarker of NHL. In addition, we provide additional evidence that supports **IA** as a potential agent for the treatment of NHL. Computational techniques such as PPI network, GO, REAC, ADMET, molecular docking, and KEGG pathway associated with TMT may be used as tools to find potential targets of drugs in NHL. Additionally, our findings could have potential clinical value in the diagnosis and therapeutic treatment of NHL beyond enriching the existing information of the pharmacological activity of IA on NHL. This study confirms that **IA** is a sesquiterpene lactone with several targets and several pathways. However, our research was limited by the lack of experimental validation, particularly in the bioinformatics approaches, and in the translatability of mouse NHL models to human disease. Moreover, clinical studies should be conducted focusing on Fhl1 as a key hub gene for early NHL diagnosis and as a potential target in precision medicine for NHL. Therefore, future studies are needed to investigate the role and impact of Fhl1 on patients with NHL and to elucidate its precise actions underlaying their effects. Furthermore, systematic studies are necessary to validate the protein targets, changes at the protein level in treated lymph nodes, and role of specific proteins in NHL. These studies should include immunohistochemistry, Western blotting, ELISA, and miRNA expression, among others.

## Figures and Tables

**Figure 1 pharmaceuticals-18-01263-f001:**
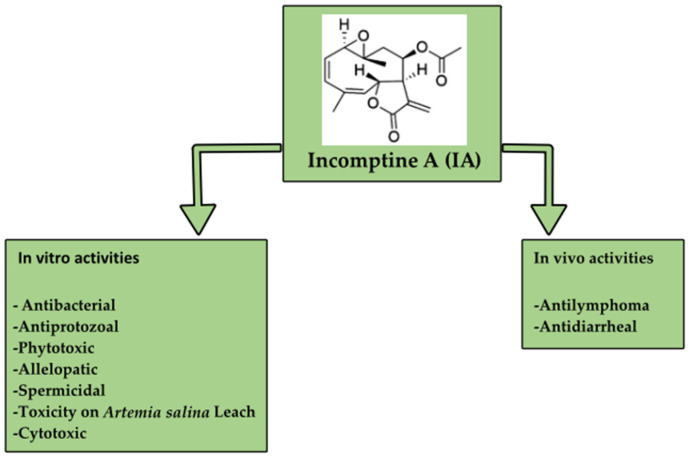
Pharmacological activities reported for incomptine A (**IA**).

**Figure 2 pharmaceuticals-18-01263-f002:**
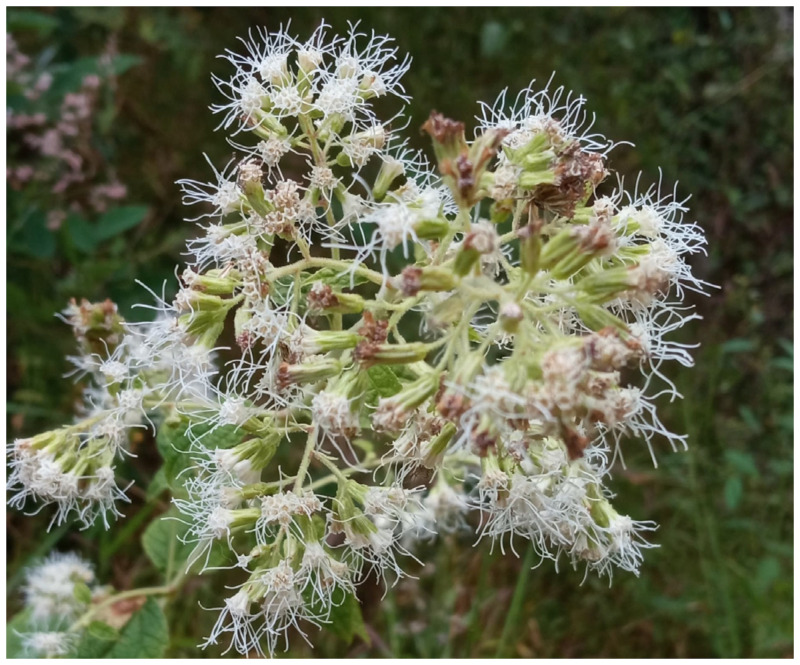
*Decachaeta incompta* (DC) R. M. King and H. Robinson.

**Figure 3 pharmaceuticals-18-01263-f003:**
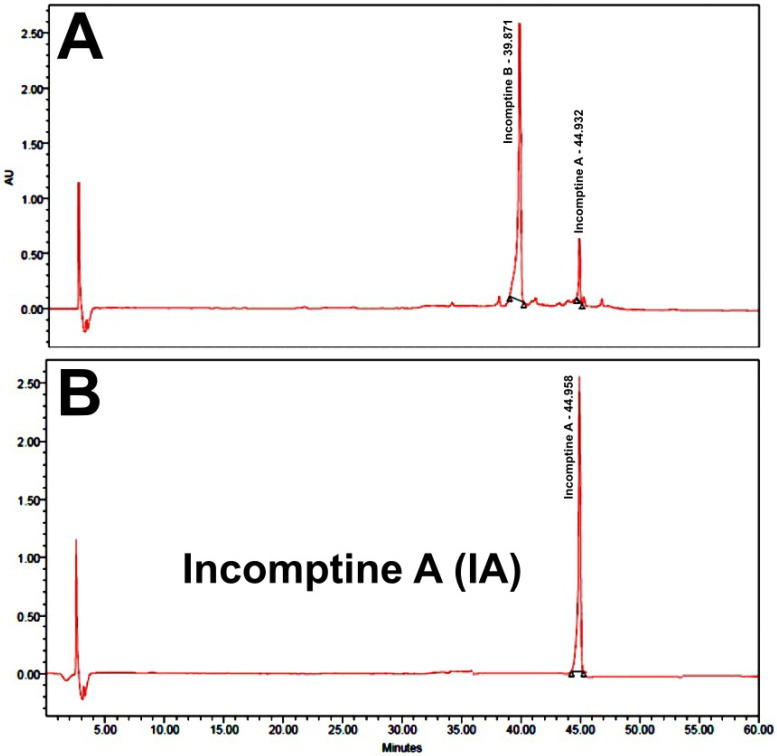
Chromatographic profile (240 nm) of dichloromethane extract from Decachaeta incompta (DCEDi) leaves (**A**) and incomptine A (**B**) standard. The x-axis indicates the retention time in minutes, while the y-axis indicates the peak % signal intensity.

**Figure 4 pharmaceuticals-18-01263-f004:**
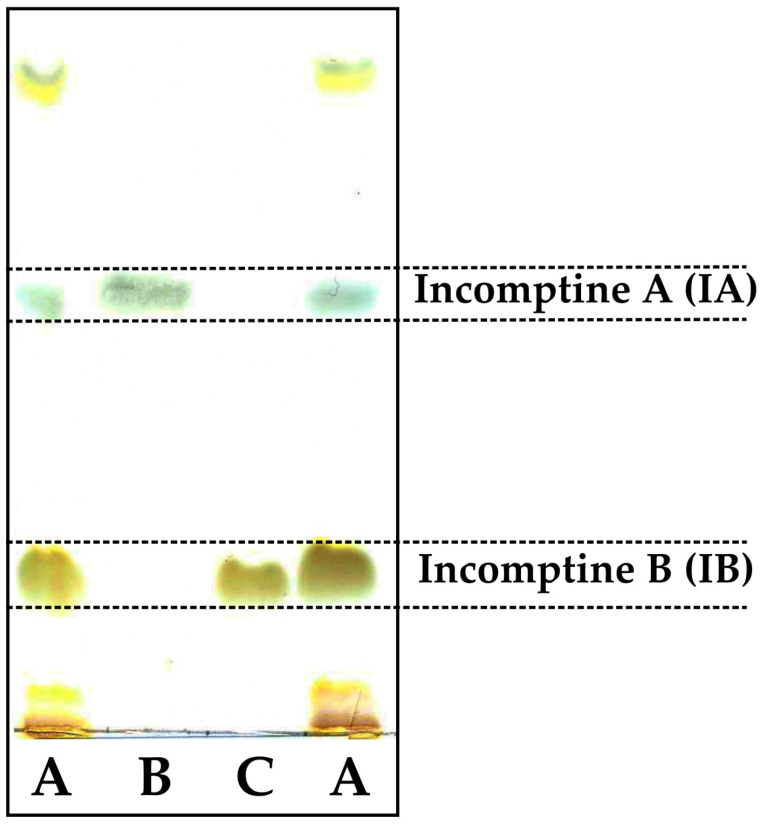
Thin-layer chromatography of presumptive identification of incomptine A (**IA**) in dichloromethane extract from *Decachaeta incompta* (DCEDi) leaves. (A) DCEDi leaves; (B) incomptine A; (C) incomptine B. Mobile phase: CHCl_3_: EtOAc (7.5: 2.5, v/v); adsorbent: silica gel 60F_254_ pre-coated TLC plates (Merck, Darmstadt); detection: compounds and extract were visualized by spraying with a 10 % solution of H_2_SO_4_ in water and then heating at 120 °C.

**Figure 5 pharmaceuticals-18-01263-f005:**
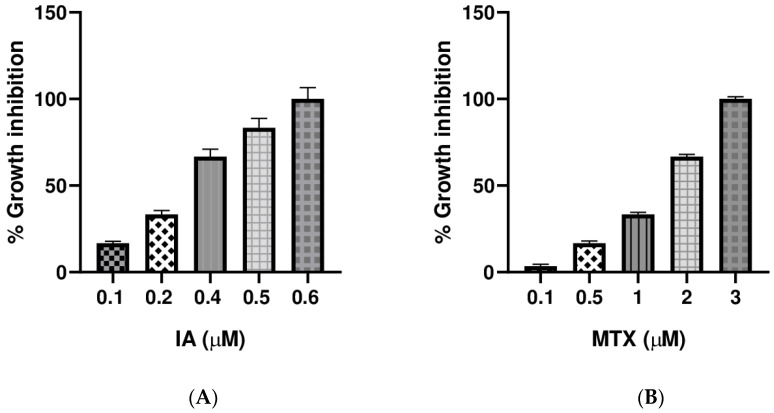
Cytotoxic activity of **IA** and **MTX** against U-937 cells (histiocytic lymphoma). The graphs show the % growth inhibition of the U-937 cell line treated with **IA** (**A**) and **MTX** (**B**) at different concentrations after 24 h of exposure. Data were analyzed using Graph Pad Prism 6.03 (*n* = 3 and *p* < 0.05, GraphPad, La Jolla, CA, USA); CC_50_ was defined as the treatment concentration at which 50% reduction in cellular proliferation was observed. The assays were performed in triplicate, and these data were used to obtain the CC_50_ by lineal regression analysis.

**Figure 6 pharmaceuticals-18-01263-f006:**
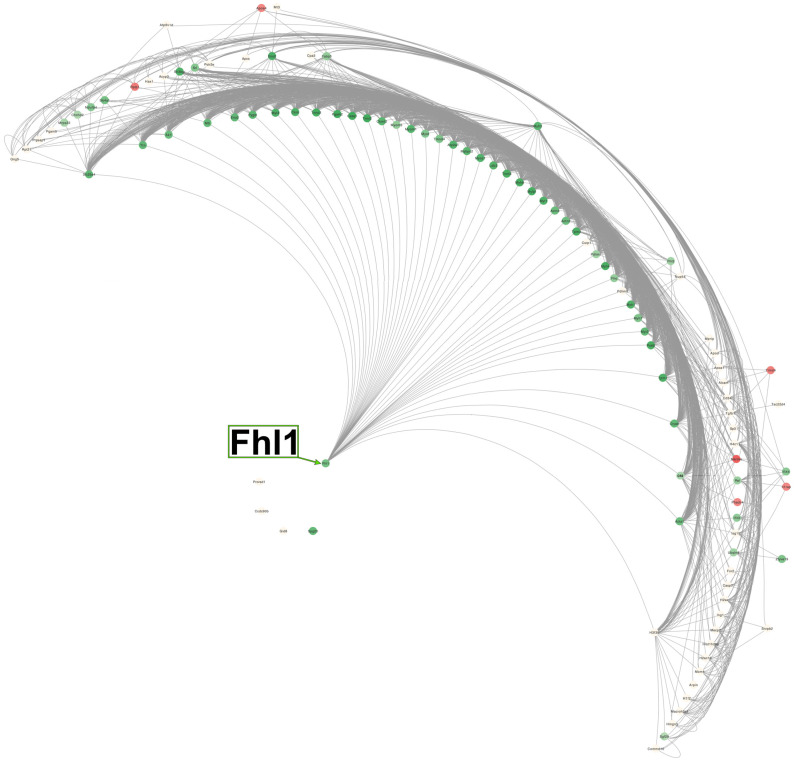
PPI network map of protein clusters and core targets in male Balb/c mice with non-Hodgkin lymphoma treated with incomptine A. The green color shows downregulated proteins and the red/cream color shows upregulated proteins. PPI, protein–protein interaction.

**Figure 7 pharmaceuticals-18-01263-f007:**
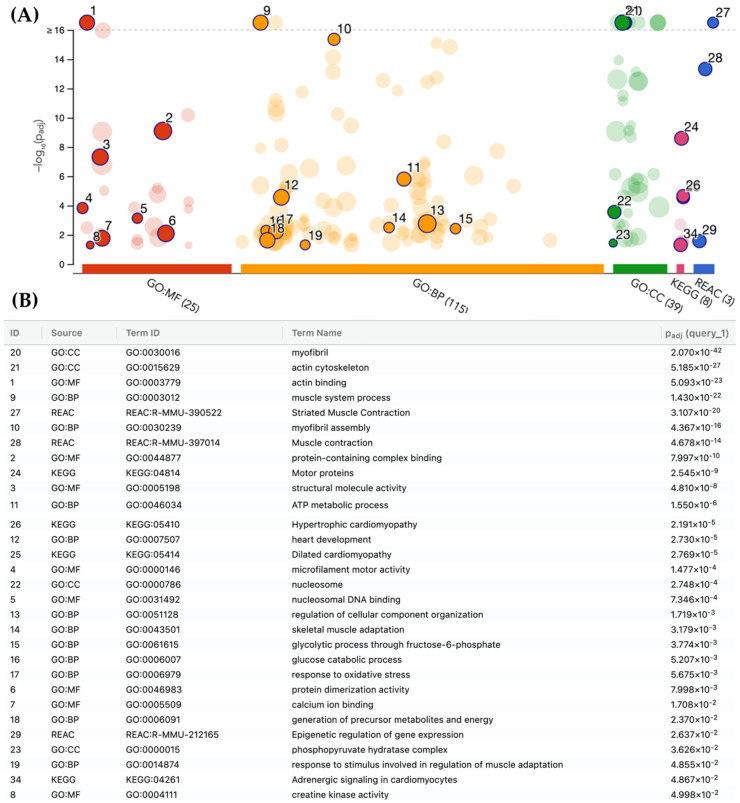
Results of bioinformatics analysis using g: Profiler. (**A**) Significant enriched terms in the GO, REAC, and KEGG databases. GO: MF, gene ontology molecular function; GO: BP, gene ontology process; GO: CC, gene ontology cellular component; KEGG, Kyoto Encyclopedia of Genes and Genomes; REAC, Reactome. (**B**) List of top 30 significantly changed terms enriched by GO, KEGG, and REAC databases.

**Figure 8 pharmaceuticals-18-01263-f008:**
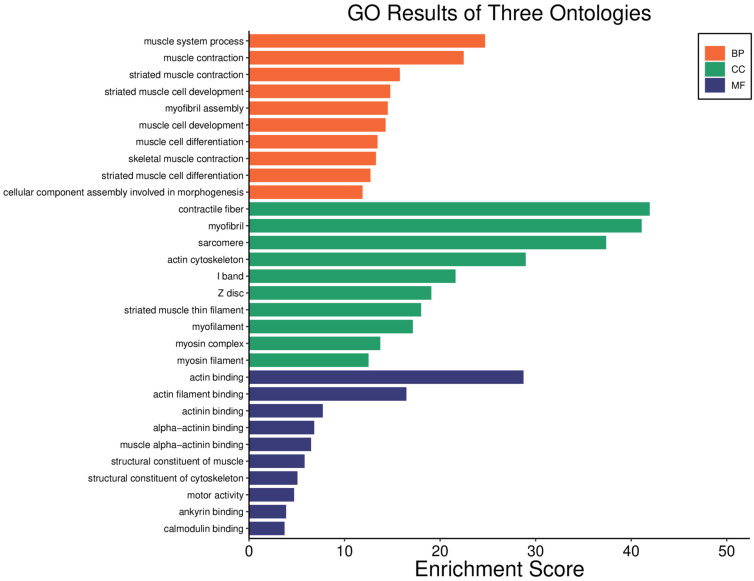
Gene ontology (GO) enrichment analysis of DEPs obtained from male Balb/c mice with non-Hodgkin lymphoma treated with incomptine A. BP, biological process; CC, cellular component; MF, molecular function.

**Figure 9 pharmaceuticals-18-01263-f009:**
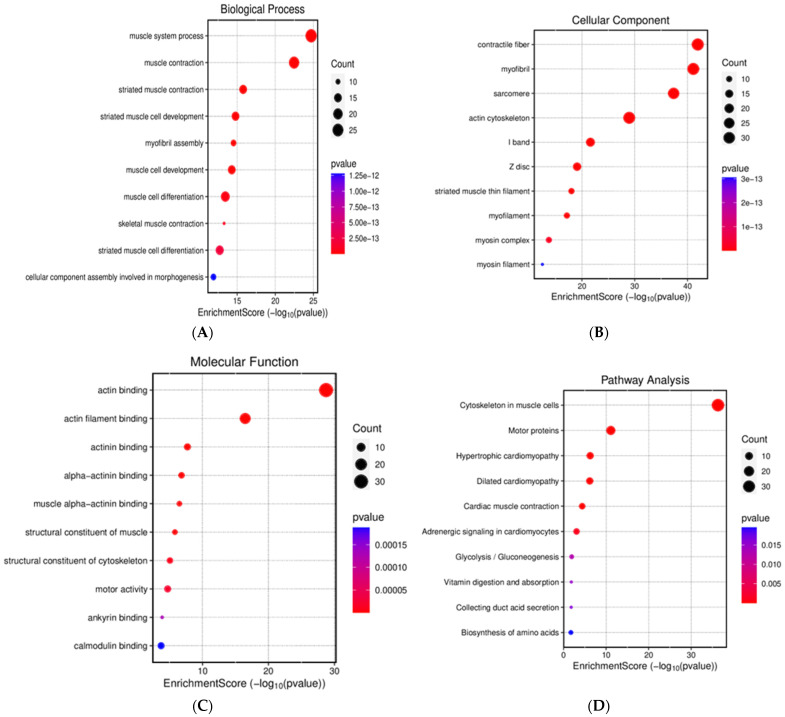
GO enrichment analysis and KEGG pathway analysis in male Balb/c mice with non-Hodgkin lymphoma. (**A**) Gene ontology (GO) in terms of biological process; (**B**) GO in terms of cellular component; (**C**) GO in terms of molecular function; (**D**) KEGG pathway analysis.

**Figure 10 pharmaceuticals-18-01263-f010:**
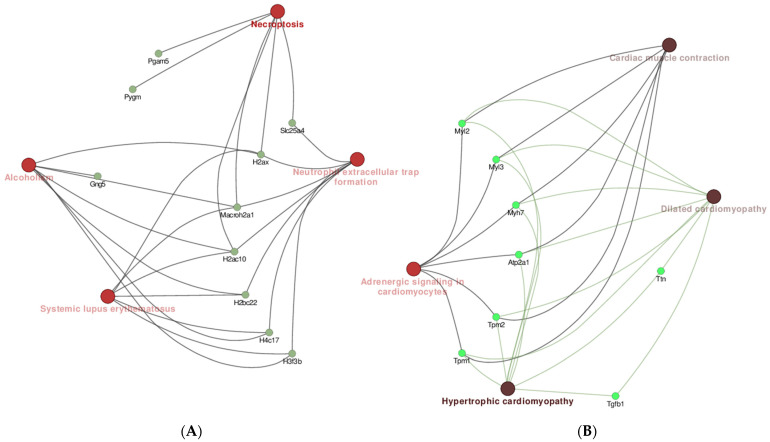
Correlation between DEPs and pathways in KEGG analysis: (**A**) necroptosis; (**B**) cardiac muscle contraction and dilated cardiomyopathy.

**Figure 11 pharmaceuticals-18-01263-f011:**
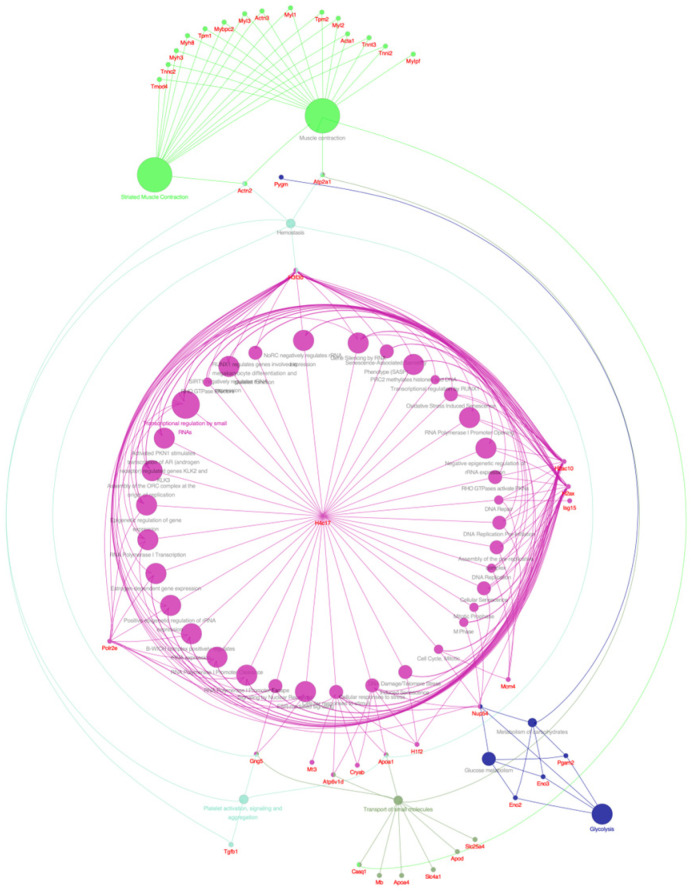
Correlation between DEPs and networks in REAC analysis.

**Figure 12 pharmaceuticals-18-01263-f012:**
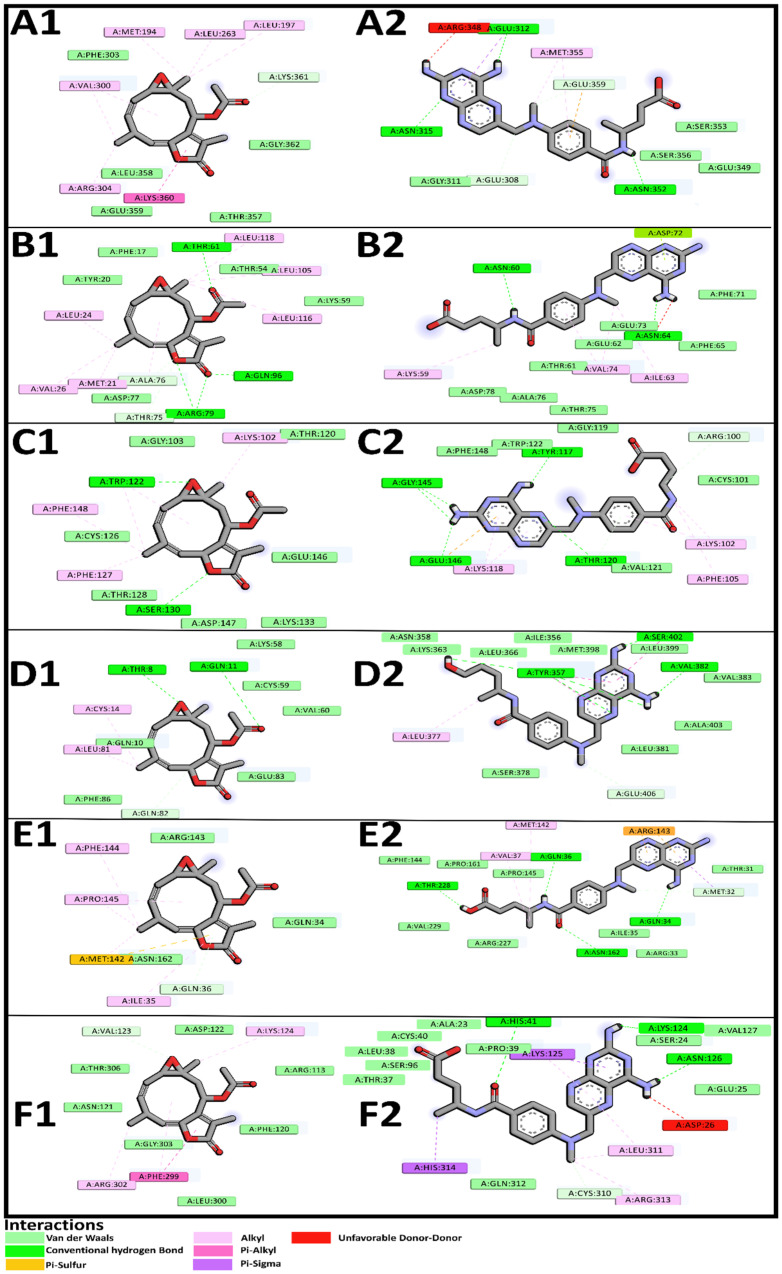
The interaction mode between incomptine A (**1**) and methotrexate (**2**) and six potential targets: (**A**) apolipoprotein A-IV, (**B**) fatty acid-binding, (**C**) four and a half LIM domains protein 1, (**D**) interferon-induced protein 44, (**E**) interleukin-1 receptor accessory protein, and (**F**) T-cell immunoglobulin and mucin domain-containing protein 4.

**Table 1 pharmaceuticals-18-01263-t001:** HPLC coupled to diode array, retention time, and percentage in area of incomptine A identified of the dichloromethane extract from Decachaeta incompta leaves.

Compound	R.T. * (min)	Area (%)	Molecular Weight (g/mol)	Molecular Formula
Incomptine A	44.95	10.42	304.35	C_17_H_20_O_5_

* R.T.: Retention time.

**Table 2 pharmaceuticals-18-01263-t002:** **^1^**H and ^13^C NMR data (500 MHz and 125 MHz) of incomptine A (**IA**) in CDCl_3_.

Position	Incomptine A	
	δ_H_, mult. (*J* in Hz)	δ_C_ (ppm)
1	3.26 dd (7.5, 1.0)	60.4
2	5.55 dd (11.5, 7.5	128.3
3	6.14 d (11.5)	132.3
4	-	136.0
5	5.31 dq (11.0, 1.5)	126.1
6	5.01 dd (11.0, 1.5)	75.7
7	2.94 quin (3.0, 1.5)	50.1
8	5.20 ddd (4.5, 3.0, 2.0)	77.0
9	2.70 dd (14.5, 4.5)	43.0
9’	1.38 ddd (14.5, 3.0, 0.5	
10	-	61.2
11	-	137.2
12	-	169.7
13	6.40 dd (2.0, 0.5)	124.8
13’	5.80 d (2.0	
14	1.41 s	19.7
15	1.88 s	23.7
Ac (C=O)	-	169.2
Ac (CH_3_)	2.02 s	20.7

**Table 3 pharmaceuticals-18-01263-t003:** Expected pharmacokinetic and toxicological properties for incomptine A (IA) and methotrexate (MTX).

Pharmacokinetics Properties						
**Absorption**				**Metabolism**		
	**IA**	**MTX**			**IA**	**MTX**
Gastrointestinal absorption	High	Low		CYP2C9 substrate	No	No
Hematoencephalic barrier	Yes	No		CYP2D6 substrate	No	No
Caco-2 permeability	High	Low		CYP3A4 substrate	Yes	Yes
p-glycoprotein substrate	Yes	Yes		CYP2C9 inhibitor	No	No
p-glycoprotein inhibitor	No	No		CYP2D6 inhibitor	No	No
Log Kp (skin permeation)	–6.89 cm/s	–10.39 cm/s		CYP3A4 inhibitor	No	No
				CYP1A2 inhibitor	No	No
				CYP2C19 inhibitor	No	No
**Distribution**				**Excretion**		
Mitochondrial	Yes	Yes		CL	4.64 mL/min/Kg	2.41 mL/min/Kg
Protein plasma binding	75.9%	63.4%		T_1/2_	0.78 h	0.39 h
Volume distribution	1.38 L/Kg	0.32 L/Kg				
**Toxicity**						
	**IA**	**MTX**			**IA**	**MTX**
Hepatotoxicity	Inactive	Active		Carcinogenicity	Inactive	Inactive
Neurotoxicity	Inactive	Active		Immunotoxicity	Inactive	Inactive
Nephrotoxicity	Inactive	Inactive		Mutagenicity	Inactive	Inactive
Respiratory toxicity	Active	Active		Cytotoxicity	Inactive	Inactive
Cardiotoxicity	Inactive	Inactive				
Predicted rats LD_50_	2.68 mol/Kg	3.49 mol/Kg				
Predicted human LD_50_	1330 mg/Kg	3.0 mg/Kg				
Expected toxicity class *	IV	I				

Predictions were obtained from ADMETlab, SwissADME, admetSAR, and PROTOX web servers. T_1/2_: Half lifetime; * Toxicity classes are defined according to the Globally Harmonized System of Classification and Labeling of Chemicals (GHS). LD_50_ is expressed in mg/kg. Class I: fatal by ingestion (LD_50_ ≤ 5); Class II: fatal by ingestion (5 < LD_50_ ≤ 50); Class III: fatal by ingestion (50 < LD_50_ ≤ 300); Class IV: fatal by ingestion (300 < LD_50_ ≤ 2000); Class V: fatal by ingestion (2000 < LD_50_ ≤ 5000); Class VI: fatal by ingestion (LD_50_ > 5000).

**Table 4 pharmaceuticals-18-01263-t004:** Physicochemical properties for **IA** and **MTX**.

**Physicochemical Properties**					
	**IA**	**MTX**	**Drug-Likeness**		
Molecular formula	C_17_H_22_O_5_	C_20_H_22_N_8_O_5_			
Molecular weight	306.35 g/mol	454.44 g/mol		**IA**	**MTX**
TPSA	65.13 Å^2^	210.54 Å^2^	Lipinsky	Yes	Yes
Lipophilicity (LogP)	2.33	−0.32	Ghose	Yes	Yes
Water solubility (LogS)	−2.71	−2.41	Veber	Yes	No, 1 violation
Solubility class	Soluble	Very soluble	Egan	Yes	No, 1 violation
Number of rotating links	2	10	Muegge	Yes	No, 1 violation
Number of H-bond donors	0	5	PAINS	0	0
Number of H-bond acceptors	5	9			

Predictions were obtained from ADMETlab and SwissADME.

**Table 5 pharmaceuticals-18-01263-t005:** Median cytotoxic concentration calculated against U-937 cell line from **IA** and **MTX**.

Compound	CC_50_ (μM) ^a^
Incomptine A (**IA**)	0.29 ± 0.01 *
Methotrexate (**MTX**)	1.60 ± 0.02

^a^ U-937 cell line (histiocytic lymphoma); CC_50_: median cytotoxic concentration causing 50% cell death. Calculated by linear regression analysis of percentage mortality against concentration. Data are expressed as mean ± SEM (*n* = 3 and * *p* < 0.05).

**Table 6 pharmaceuticals-18-01263-t006:** Results of the antilymphoma activity after the treatment with **IA** and **MTX** on male Balb/c mice with NHL.

Compound	% Inhibition (10 mg/kg) in Male Balb/c Mice ^a^	ED_50_ (mg/kg) ^b^
Incomptine A (**IA**)	62.0 ± 1.5	7.5 ± 0.01
Methotrexate (**MTX**)	ND *	1.4 ± 0.02

^a^ Results are expressed as average ± SEM (*n* = 6 and *p* < 0.05). * ND: not determined; MTX at doses ≥ 1.25 mg/kg caused mortality in male Balb/c mice. ^b^ ED50: median effective dose that caused 50% of mice to have the desired pharmacological effect.

**Table 7 pharmaceuticals-18-01263-t007:** Differentially expressed proteins for the treatment of **IA** or **MTX** in lymph nodes from male Balb/c mice with non-Hodgkin lymphoma.

C-(DMSO) vs	Downregulated FC < 0.66 (1/1.5)	Upregulated FC > 1.5
MTX	57	7
IA	66	40

Data represents the fold change (FC) with the number of DEPs. **IA** at a dose of 10 mg/kg, **MTX** at a dose of 1.25 mg/kg and negative control, male Balb/c mice with NHL. DEPs were compared with mouse protein.

**Table 8 pharmaceuticals-18-01263-t008:** Proteins altered for **IA** only (10 mg/kg) in lymph nodes from male mice with NHL and treated with **IA** or **MTX** (1.25 mg/kg) and identified by TMT.

Protein ID	Protein Name	Gene Name	IA (FC)	MTX (FC)
P04202	Transforming growth factor beta 1	Tgfb1	0.643493	WC
P97434	Myosin phosphatase Rho interacting protein	Mprip	0.637507	WC
Q9EQN3	Domain family member 4	Tsc22d4	0.621895	WC
P49717	DNA replication licensing factor MCM4	Mcm4	0.605720	WC
P56375	Acylphosphatase-2	Acyp2	0.612446	WC
Q64339	Ubiquitin-like protein ISG15	Isg15	0.596993	WC
Q8CI51	PDZ and LIM domain protein 5	Pdlim5	0.610126	WC
Q9QXV3	Inhibitor of growth protein 1	Ing1	0.612365	WC
Q9WVR4	RNA-binding protein	Fxr2	0.604267	WC
P15089	Mast cell carboxypeptidase A	Cpa3	1.684613	WC
P57746	V-type proton ATPase subunit D	Atp6v1d	1.692460	WC
Q00623	Apolipoprotein A-I	Apoa1	1.714051	WC
Q61490	CD 106 antigen	Alcam	1.592873	WC
Q64314	Hematopoietic progenitor cell antigen CD34	Cd34	1.557006	WC
Q80SZ7	Guanine nucleotide-binding protein G(I)/G(S)/G(O)	Gng5	1.847469	WC
Q8C3X2	Coiled-coil domain-containing protein 90B	Ccdc90b	1.602073	WC
Q9CQI7	U2 small nuclear ribonucleoprotein B	Snrpb2	1.583845	WC
Q9D0M1	Phosphoribosyl pyrophosphate synthase-associated protein 1	Prpsap1	1.605011	WC
Q9D7M1	Glucose-induced degradation protein 8 homolog	Gid8	1.630454	WC
Q9JL35	High mobility group nucleosome-binding domain containing protein 5	Hmgn5	1.665424	WC
O09167	Large ribosomal subunit protein eL21	Rpl21	2.107700	WC
O35387	HCLS1-associated protein X-1	Hax1	1.777345	WC
O70494	Transcription factor Sp3	Sp3	1.687230	WC
P84244	Histone H3.3	H3f3a; H3f3b	2.804078	WC
Q8CGP2	Histone H2B type 1-P	Hist1h2bp	2.244255	WC
P12246	Serum amyloid P-component	Apcs	1.636670	WC
P15864	Histone H1.2	H1-2 or Hist1h1c	2.578941	WC
P27661	Histone H2AX	H2ax or H2afx	2.641841	WC
P28184	Metallothionein-3	Mt3	1.630307	WC
P51910	Apolipoprotein D	Apod	1.835178	WC
P62806	Histone H4	H4c11 or Hist1h4a	3.246594	WC
P97315	Cysteine and glycine-rich protein 1	Csrp1	1.620372	WC
P97864	Caspase-7	Casp7	1.611143	WC
Q80UW8	DNA-directed RNA polymerases I, II, and III subunit RPABC1	Polr2e	1.715163	WC
Q8BTS4	Nuclear pore complex protein Nup54	Nup54	1.595948	WC
Q8BX10	Serine/threonine-protein phosphatase PGAM5, mitochondrial	Pgam5	1.578568	WC
Q8CGP5	Histone H2A type 1-F	Hist1h2af	2.449976	WC
Q8JZY2	COMM domain-containing protein 10	Commd10	1.674767	WC
Q9D0A3	Arpin	Arpin	1.560642	WC
Q9D820	Prolyl-tRNA synthetase associated domain-containing protein 1	Prorsd1	1.790119	WC
Q9QZQ8	Core histone macro-H2A.1	Macroh2a1 or H2afy	2.275530	WC
Q9Z2D6	Methyl-CpG-binding protein 2	Mecp2	1.942897	WC

FC, fold change; WC, without change.

**Table 9 pharmaceuticals-18-01263-t009:** Proteins altered by **IA** (10 mg/kg) and **MTX** (1.25 mg/kg) in lymph nodes from male mice with NHL and identified by TMT.

Protein ID	Protein Name	Gene Name	IA (FC)	MTX (FC)
O09165	Calsequestrin-1	Casq1	0.116192	0.168875
O70250	Phosphoglycerate mutase 2	Pgam2	0.130508	0.153858
P13412	Troponin I, fast skeletal muscle	Tnni2	0.124295	0.155632
P20801	Troponin C, skeletal muscle	Tnnc2	0.138595	0.178338
P17183	Gamma-enolase	Eno2	0.151492	0.204480
Q9QZ47	Troponin T, fast skeletal muscle	Tnnt3	0.168688	0.187661
P97457	Myosin regulatory light chain 2, skeletal muscle isoform	Mylpf	0.160783	0.185690
P13542	Myosin-8	Myh8	0.167594	0.188003
Q9WUZ7	SH3 domain-binding glutamic acid-rich protein	Sh3bgr	0.181164	0.214274
P05977	Myosin light chain 1/3, skeletal muscle isoform	Myl1	0.180140	0.223749
Q5SX39	Myosin-4	Myh4	0.188800	0.191696
P58774	Tropomyosin beta chain	Tpm2	0.183496	0.197290
P32848	Parvalbumin alpha	Pvalb	0.186871	0.197277
Q5SX40	Myosin-1	Myh1	0.193077	0.224988
Q9JKS4	LIM domain-binding protein 3	Ldb3	0.210207	0.245489
P07310	Creatine kinase M-type	Ckm	0.205044	0.218269
P58771	Tropomyosin alpha-1 chain	Tpm1	0.211997	0.215467
Q9JK37	Myozenin-1	Myoz1	0.212286	0.240133
Q8R429	Sarcoplasmic/endoplasmic reticulum calcium ATPase 1	Atp2a1	0.241364	0.246236
Q62234	Myomesin-1	Myom1	0.235461	0.252929
Q6P8J7	Creatine kinase S-type, mitochondrial	Ckmt2	0.237430	0.265434
Q8R1X6	Spartin	Spg20	0.255282	0.309371
P51667	Myosin regulatory light chain 2, ventricular/cardiac	My12	0.244251	0.292434
	muscle isoform			
P09542	Myosin light chain 3	My13	0.238113	0.2115191
Q5XKE0	Myosin-binding protein C, fast-type	Mybpc2	0.257520	0.274625
Q9WUB3	Glycogen phosphorylase, muscle form	Pygm	0.250124	0.280577
P13541	Myosin-3	Myh3	0.245456	0.259989
A2ASS6	Titin	Ttn	0.278499	0.310760
P21550	Beta-enolase	Eno3	0.287361	0.301657
P23927	Alpha-crystallin B chain	Cryab	0.280105	0.307071
Q9JI91	Alpha-actinin-2	Actn2	0.282126	0.349733
O88990	Alpha-actinin-3	Actn3	0.309565	0.335477
P04247	Myoglobin	Mb	0.327717	0.314839
Q9JLH8	Tropomodulin-4	Tmod4	0.342717	0.419507
P97447	Four and a half LIM domains protein 1	Fhl1	0.359150	0.385025
P11404	Fatty acid-binding protein, heart	Fabp3	0.375626	0.505352
P68134	Actin, alpha skeletal muscle	Acta1	0.401163	0.351810
Q9JIF9	Myotilin	Myot	0.393725	0.442669
P48962	ADP/ATP translocase 1	Slc25a4	0.434997	0.292006
Q9DAZ9	Abscission/NoCut checkpoint regulator	Zfyve19	0.463985	0.424291
Q64345	Interferon-induced protein with tetratricopeptide repeats 3	Ifit3	0.408482	0.500078
Q9R0Y5	Adenylate kinase isoenzyme 1	Ak1	0.394418	0.342173
Q7TQ48	Sarcalumenin	Srl	0.417223	0.452027
Q3TJD7	PDZ and LIM domain protein 7	Pdlim7	0.423444	0.581780
Q8VHX6	Filamin-C	Flnc	0.460515	0.528664
Q91Z83	Myosin-7	Myh7	0.450777	0.432883
A2ABU4	Myomesin-3	Myom3	0.447776	0.505737
Q8BV66	Interferon-induced protein 44	Ifi44	0.549142	0.481085
P15307	Proto-oncogene c-Rel	Rel	0.513074	0.548115
P04919	Band 3 anion transport protein	Slc4a1	0.506732	0.390399
Q99NB8	Ubiquilin-4	Ubqln4	0.453602	0.576117
Q9CQC7	NADH dehydrogenase [ubiquinone] 1 beta subcomplex	Ndufb4	0.581219	0.459827
	subunit 4			
P45591	Cofilin-2	Cfl2	0.597159	0.624516
Q9DA08	SAGA-associated factor 29 homolog	Ccdc101	0.594209	0.656104
Q9D1L0	Coiled-coil-helix-coiled-coil-helix domain-containing protein 2	Chchd2	0.580712	0.635534
Q9CXW2	28S ribosomal protein S22, mitochondrial	Mrps22	0.635789	0.602915
Q9R059	Four and a half LIM domains protein 3	Fhl3	0.645077	0.581828
P06728	Apolipoprotein A-IV	Apoa4	1.522175	1.508959
P01645	Ig kappa chain V-V region HP 93G7	KV5AC	1.748303	1.688753
P47955	Large ribosomal subunit protein P1	Rplp1	1.574290	1.558623
Q501J7	Phosphatase and actin regulator 4	Phactr4	1.603299	1.505982
Q6U7R4	T-cell immunoglobulin and mucin domain-containing protein 4	Timd4	1.725275	1.539719
Q61730	Interleukin-1 receptor accessory protein	Il1rap	1.665689	1.502210
P26645	Myristoylated alanine-rich C-kinase substrate	Marcks	1.891390	1.679438

FC, fold change.

**Table 10 pharmaceuticals-18-01263-t010:** Protein–ligand interactions from molecular docking, ∆G (kcal/mol), and Ki (µM).

**Protein**	**Ligands**								
	**Incomptine A (IA)**					**Methotrexate (M)**			
	**ΔG**	**Ki**	**H-BR**	**NPI**		**ΔG**	**Ki**	**H-BR**	**NPI**
Apoa4	−6.35	22.12	Phe303, Thr357, Leu358, Glu359, Lys361, Gly361	Met194, Leu197, Leu263, Val300, Arg304, Lys360		−4.27	744.4	Glo308, Gly311, Glu312, Asn315, Glu349, Asn352, Ser353, Ser356, Glu359	Arg348, Met355
Fabp3	−7.17	5.53	Phe17, Tyr20, Thr54, Lys59, Thr61, Thr75, Ala76, Asp77, Arg79, Gln96	Met21, Leu24, Val26, Leu105, Leu116, Leu118		−4.84	284.07	Asn60, Thr61, Glu62, Asn64, Phe65, Phe71, Asp72, Glu73, Thr75, Ala76, Asp78	Lys59, Ile63, Val74
Fhl1	−6.86	9.38	Gly103, Thr120, Trp122, Cys126, Thr128, Ser130, Lys133, Glu146, Asp147	Lys102, Phe127, Phe148		−4.92	284.1	Arg100, Cys101, Tyr117, Gly119, Thr120, Val121, Trp122, Gly145, Glu146, Phe148	Lys102, Phe105, Lys118
IFi44	−6.55	15.9	Thr8, Gln10, Gln11, Lys58, Cys59, Val60, Gln82, Glu83, Phe86	Cys14, Leu81		−5.72	64.22	Ile356, Tyr357, Asn358, Lys363, Leu366, Ser378, Leu381, Val382, Val383, Met398, Leu399, Ser402, Ala403, Glu406	Leu377
Il1rap	−7.15	5.74	Gln34, Gln36, Arg143, Arg143	Ile35, Met142, Phe144, Pro145		−6.02	38.51	Thr31, Met32, Arg33, Gln34, Ile35, Gln36, Phe144, Pro145, Pro161, Asn162, Arg227, Thr228, Val229	Val37, Met142, Arg143
Timd4	−7.24	4.9	Arg113, Phe120, Asn121, Asp122, Val123, Leu300, Gly303, Thr306	Lys124, Phe299, Arg302		−5.11	180.54	Ala23, Ser24, Glu25, Thr37, Leu38, Pro39, Cys40, His41, Ser96, Ala124, Asn126, Val127, Cys310, Gln312,	Asp26, Lys215, Leu311, Arg313, His314

Apoa4: apolipoprotein A-IV, Fabp3: fatty acid-binding, Fhl1: four and a half LIM domains protein 1, Ifi44: interferon-induced protein 44, Il1rap: interleukin-1 receptor accessory protein, Timd4: T-cell immunoglobulin and mucin domain-containing protein 4.

## Data Availability

The original contributions presented in this study are included in the article. Further inquiries can be directed to the corresponding author(s).

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
