# Peer review of "Expanding the Knowledge of the Molecular Effects and Therapeutic Potential of Incomptine A for the Treatment of Non-Hodgkin Lymphoma: In Vivo and Bioinformatics Studies, Part III"

_pharmaceuticals, 2025, doi:10.3390/ph18091263_

Round 1

Reviewer 1 Report

Comments and Suggestions for Authors

Thank you for the opportunity to review this manuscript. As a reviewer, my general overview of this manuscript suggests that IA has significant antilymphoma activity, favourable pharmacokinetics, lower toxicity, and modulates numerous NHL-relevant proteins and pathways. This study is scientifically robust and methodologically thorough, but would greatly benefit from targeted experimental validation and expanded toxicity profiling. Furthermore, to strengthen the manuscript further and to further expose the mechanistic and translational potential of IA, I suggest the following additional potential experiments.

Comments:

  1. The author should consider those experiments that had validated protein targets, either Western blotting or ELISA for specific DEPs (i.e., Fhl1, Il1rap, Fabp3, IFi44, etc), to validate changes at the protein level in treated lymph nodes.
  2. Secondly, I have concerns related to functional assays that are relevant to this study, for example, Apoptosis assays (i.e., caspase activity, TUNEL staining, Annexin V/PI flow cytometry) to show whether IA promotes lymphoma cell death via apoptosis or some other mechanism.
  3. If the author were able to knock down the key targets or overexpression of targets to examine resistance to IA, this study would be on the next level.
  4. For potential therapeutic drug discovery in this pathway, we never ignore the critical part, which is the In Vivo Toxicity of that compound. I encourage the author to perform in vivo toxicity assessments on healthy mice, including liver, kidney, and neurological evaluation, to confirm all the predicted findings in the manuscript as outlined computationally.
  5. The author needs to provide information on statistical methods, p-values, and software parameters selected for pathway enrichment and differential expression.
  6. Figure legends require improvement, and graphical abstracts are created for key pathways and molecular docking studies.
  7. The author needs to acknowledge the potential limitations and translatability of mouse NHL models to human disease.

Author Response

August 15, 2025

Dear, Reviewer 1

The manuscript ID: pharmaceuticals-3815701; Type of manuscript: Article
Title: Expanding the Know of Molecular Effects and Therapeutic Potential of Incomptine A for the Treatment of Non-Hodgkin Lymphoma: In Vivo and Bioinformatics Studies, Part III; was reviewed in aggrement with the comments and suggestions of reviewers 1 and 2. Additional texts were included in yellow color..

Fernando Calzada

PhD Profesor

Comments and Suggestions for Authors

Thank you for the opportunity to review this manuscript. As a reviewer, my general overview of this manuscript suggests that IA has significant antilymphoma activity, favourable pharmacokinetics, lower toxicity, and modulates numerous NHL-relevant proteins and pathways. This study is scientifically robust and methodologically thorough, but would greatly benefit from targeted experimental validation and expanded toxicity profiling. Furthermore, to strengthen the manuscript further and to further expose the mechanistic and translational potential of IA, I suggest the following additional potential experiments.

ANSWER:

Dear, reviewer thanks for your important comments and suggestion for the enrichment of this work.

Comments:

  1. The author should consider those experiments that had validated protein targets, either Western blotting or ELISA for specific DEPs (i.e., Fhl1, Il1rap, Fabp3, IFi44, etc), to validate changes at the protein level in treated lymph nodes.

ANSWER:

In realtion to the comments 1. We are aggrement with these suggestion. However, in this moment for the time to perform and cost of experiments is very difficult for our laboratory have in this year. In this sense additional text was include in Discussion and Conclusions sections; lines 551 to 553 and 786 to 789, respectively.

Additional assays including immunohistochemistry, Western blotting, ELISA, and miRNA expression among others are mandatory to validate the results obtained here.

Also, systematic studies are necessary to validate the protein targets, changes at the protein level in treated lymph nodes and role of specific proteins in NHL. These studies including immunohistochemistry, Western blotting, ELISA, and miRNA expression among others.  

  1. Secondly, I have concerns related to functional assays that are relevant to this study, for example, Apoptosis assays (i.e., caspase activity, TUNEL staining, Annexin V/PI flow cytometry) to show whether IA promotes lymphoma cell death via apoptosis or some other mechanism.

ANSWER:

In realtion to the comments 2. We are aggrement with these suggestion. In this sense for the moment considering the time to perform and cost of experiments is very difficult for our laboratory have these suggestions in this year. I additional text was include in Discussion section, lines 551 to 561.

Also, others assays are required to explore more molecular mechanisms of as IA promotes lymphoma cell death in NHL such as apoptosis (i.e., caspase activity, TUNEL staining, and annexin V/PI flow cytometry) and effect on glycolytic pathway (i.e., expression of aldolase A, lactate dehydrogenase, and hexokinase II).   

  1. If the author were able to knock down the key targets or overexpression of targets to examine resistance to IA, this study would be on the next level.

ANSWER:

At the momento we do not have those experiments. This suggestion will be considered to future research.

  1. For potential therapeutic drug discovery in this pathway, we never ignore the critical part, which is the In Vivo Toxicity of that compound. I encourage the author to perform in vivo toxicity assessments on healthy mice, including liver, kidney, and neurological evaluation, to confirm all the predicted findings in the manuscript as outlined computationally.

ANSWER:

Additional text was included about in vivo toxicity of IA reported previousl. Lines 420 to 423.

It is important to highlight that the result of toxicity obtained in this study to IA, was in agreement with the in vivo toxicity reported previously and confirms the predicted findings by bioinformatic

  1. The author needs to provide information on statistical methods, p-values, and software parameters selected for pathway enrichment and differential expression.

ANSWER:

Additional text was included to Antilymphoma Test (lines 635 to 639), Cytotoxic Activity  (lines 607 to 610), pathways enrichment (lines 699 to 707) and differential expression (lines 676 to 679; 683 to 686).

Statistical analysis of the data was performed using one-way ANOVA, as well as Dunnett´s multiple comparison tests with a value of p < 0.05 to establish a significant difference between the study groups.

Dotplot and three onthologies plots enrichment was performed with the ClusterProfiler package of the R v4.2.2 tool, following the steps proposed by the developer, using the GO databases (BP, MF, CC) and FDR < 0.05 as the value of statistical significance. Additionally, enrichment networks were performed in Cytoscape using the ClueGo as well as STRING plugins (confidence cutoff of 0.4), for both analyses, following the developer's manual using KEGG and GO databases and setting a p < 0.05 as the statistical significance criterion.  In parallel, we performed a functional enrichment analysis, also known as over-representation analysis (ORA), using the gProfiler tool available online, for which we used the parameters preset by the program

Data analytical report were carried out by Analytical Service from Creative Proteomics (New York, NY, USA) with p < 0.05 (indicates > 95% confidence of change in protein concentration respect of the magnitude of change) selected to designate differentially expressed proteins.

 It is important to highlight that fold change cutoffs at protein level accepted with biological meaning are defined as 1.5 times up or down. Therefore, the screening criteria to the relative quantitation of proteins in this work

  1. Figure legends require improvement, and graphical abstracts are created for key pathways and molecular docking studies.

ANSWER:

Figure legends of 6 to 12 were improved. Also a graphical abstract is icluded

  1. The author needs to acknowledge the potential limitations and translatability of mouse NHL models to human disease.

ANSWER:

Additional text was included in conclusions

However, our research was limited by the lack of experimental validation particularly in the bioinformatics approaches. Also, in the translatability of mouse NHL models to human disease. Moreover, clinical studies focusing on Fhl1 as a key hub gene for early NHL diagnosis and as a potential target in precision medicine for NHL. Therefore, future studies are needed to investigate the role and impact of Fhl1 on patients with NHL and to elucidate its precise underlaying mechanisms

Reviewer 2 Report

Comments and Suggestions for Authors

Hi dear Editorial board and the respected authors

This article " Expanding the Know of Molecular Effects and Therapeutic Potential of Incomptine A for the Treatment of Non-Hodgkin Lymphoma: In Vivo and Bioinformatics Studies, Part III” was revised and has a novelty and I recommend it for publication after consideration of the following comments.

The study evaluates the antilymphoma properties of incomptine A (IA) in treating Non-Hodgkin lymphoma (NHL) using various methods. Results show strong binding affinities with cancer-associated proteins, positive pharmacokinetic properties, and no toxicity, suggesting IA may be a potential therapeutic agent.

Abstract:

  • The type of statistical design used in this research should be mentioned.
  • How were the Tandem Mass Tag (TMT) analysis and subsequent pathway enrichment studies used to clarify the precise molecular mechanisms underlying incomptine A's (IA) anti-lymphoma properties?
  • Could you provide more details about the biological significance of the 106 proteins that were found to be differentially expressed in the lymph nodes of Balb/c mice? Specifically, how do these proteins relate to the therapeutic effects of IA and what part do they play in the pathophysiology of non-Hodgkin lymphoma?
  • What particular techniques were employed to evaluate IA's toxicity, and how do the results lend credence to its possible application as a therapeutic agent in NHL, particularly with regard to its pharmacokinetic characteristics and interactions with cancer-related proteins?

Introduction:

  • Mechanistic Understanding: What specific pathways underlie incomptine A's (IA) cytotoxic effects against various cancer cell types, and how does IA affect NF-kB expression and apoptosis in Non-Hodgkin lymphoma cells?
  • In light of its documented cytotoxicity and low toxicity, how effective is IA in comparison to more conventional therapies for Non-Hodgkin lymphoma, such as methotrexate, in terms of therapeutic results and adverse effect profiles?
  • Proteomic and Pharmacological Insights: How might the proteomic and network pharmacology methods used in this study help guide future NHL research and treatment plans? What new biomarkers and therapeutic targets were discovered?

Materials:

  • Please write materials as Company Name (City, Country), especially for chemical analysis assessment which used in the study.

Methodology:

  • Please cite and have template “https://doi.org/10.1111/jfpp.16028 “for statistical analysis which is lack in materials and methods.

Results:

  • All Tables and Figures: The alphabetical statistical letters for the means should all be modified such that the greatest number has the letter a and as the numbers go lower, letters b, c
  • Interpretation of Pathway Analysis: What role do the enriched pathways—such as muscle contraction and glycolysis—found in the Gene Ontology and KEGG analyses play in understanding how incomptine A works to treat non-Hodgkin lymphoma, and what does this mean for potential future treatment approaches?
  • Molecular Docking Results: Could incomptine A (IA) be a new treatment option for NHL? What are the implications of molecular docking studies that demonstrate IA has a higher binding affinity to differentially expressed proteins than methotrexate?
  • Functional Validation: How will the down-regulated genes found in the Reactome analysis be validated experimentally, and how will these results influence the therapeutic potential of incomptine A in clinical settings?

Discussion:

  • Discussion text must grammar improve and in some cases it is very weak and maybe there is no discussion at all.
  • Given its lower toxicity and superior ADMET profiles, how exactly does incomptine A (IA) treat non-Hodgkin lymphoma more effectively than methotrexate (MTX)?
  • What more research is required to examine the potential of "Four and a half LIM domains protein 1" (Fhl1) as a therapeutic target and prognostic biomarker for NHL, and how might this impact future drug development strategies, considering the study's identification of its significant role?
  • How can we better understand the biological mechanisms underlying NHL by combining proteomic, bioinformatics, and molecular docking studies? What does this mean for the development of novel therapeutic agents like IA?

Conclusions:

  • Conclusion is very general, try to make it more scientific, comprehensive and concise in detail, especially.

References: It is OK.

The article has many flaws in express and concept of English, it is suggested to be revised in a scientific and native way.

Author Response

August 15, 2025

Dear, Reviewer 2

The manuscript ID: pharmaceuticals-3815701; Type of manuscript: Article
Title: Expanding the Know of Molecular Effects and Therapeutic Potential of Incomptine A for the Treatment of Non-Hodgkin Lymphoma: In Vivo and Bioinformatics Studies, Part III; was reviewed in aggrement with the comments and suggestions of reviewers 1 and 2. Additional texts were included in yellow color.

Fernando Calzada

PhD Profesor

Comments and Suggestions for Authors

Hi dear Editorial board and the respected authors

This article " Expanding the Know of Molecular Effects and Therapeutic Potential of Incomptine A for the Treatment of Non-Hodgkin Lymphoma: In Vivo and Bioinformatics Studies, Part III” was revised and has a novelty and I recommend it for publication after consideration of the following comments.

The study evaluates the antilymphoma properties of incomptine A (IA) in treating Non-Hodgkin lymphoma (NHL) using various methods. Results show strong binding affinities with cancer-associated proteins, positive pharmacokinetic properties, and no toxicity, suggesting IA may be a potential therapeutic agent.

Abstract:

  • The type of statistical design used in this research should be mentioned.

ANSWER:

Additional text was included to Antilymphoma Test (lines 635 to 639), Cytotoxic Activity  (lines 607 to 610), pathways enrichment (lines 699 to 707) and differential expression (lines 676 to 679; 683 to 686).

Statistical analysis of the data was performed using one-way ANOVA, as well as Dunnett´s multiple comparison tests with a value of p < 0.05 to establish a significant difference between the study groups.

Dotplot and three onthologies plots enrichment was performed with the ClusterProfiler package of the R v4.2.2 tool, following the steps proposed by the developer, using the GO databases (BP, MF, CC) and FDR < 0.05 as the value of statistical significance. Additionally, enrichment networks were performed in Cytoscape using the ClueGo as well as STRING plugins (confidence cutoff of 0.4), for both analyses, following the developer's manual using KEGG and GO databases and setting a p < 0.05 as the statistical significance criterion.  In parallel, we performed a functional enrichment analysis, also known as over-representation analysis (ORA), using the gProfiler tool available online, for which we used the parameters preset by the program

Data analytical report were carried out by Analytical Service from Creative Proteomics (New York, NY, USA) with p < 0.05 (indicates > 95% confidence of change in protein concentration respect of the magnitude of change) selected to designate differentially expressed proteins.

 It is important to highlight that fold change cutoffs at protein level accepted with biological meaning are defined as 1.5 times up or down. Therefore, the screening criteria to the relative quantitation of proteins in this work

  • How were the Tandem Mass Tag (TMT) analysis and subsequent pathway enrichment studies used to clarify the precise molecular mechanisms underlying incomptine A's (IA) anti-lymphoma properties?

ANSWER: All these information was include in discussion section

A total of 2717 proteins were identified and quantified  by TMT análisis in the LNNHLTIA in this study, among which 106 proteins showed significant expression differences (Table 8 and Table 9), including 66 down regulated proteins and 40 up regulated proteins. After All DEPs altered by IA treatment were subjects to PPI analysis, showing that “Four and a half LIM domains protein 1” (Fhl1) was the most important core protein (Figure 6), it showed directly or indirectly interaction with others proteins including, Il1rap, Ifi44, Timd4, Apoa4, and Fabp3 as well as Myh3, Eno 2, and H4c11, the last three was reported recently could be associated with the appearance of NHL. “Four and a half LIM domains protein 1” (Fhl1) result the most important cluster altered and potential core target of IA for the treatment of NHL in male Balb/c mice.

According to the PPI network, GO, KEGG, and REAC analyses, the DEPs including Fhl1, Il1rap, Ifi44, Timd4, Apoa4, Fabp3, Myh3, Eno 2, and H4c11, may be associated with several processes containing muscle contraction, glycolysis, hemostasis, epigenetic regulation of gene expression, transport of small molecules, neutrophil extracellular trap formation, adrenergic signaling in cardiomyocytes, systemic lupus erythematosus, alcoholism, platelet activation, signaling and aggregation (Figures 7-11).

In tumors, the expression of Fhl1 is up regulated or down regulated and plays a role in promoting or inhibiting tumor development. The expression of Fhl1 is down regulated in several cancers such as lung, prostate, breast, ovarian, colon, thyroid, brain, kidney, liver, and melanoma, as well as oral cancers. In contrast, the expression of Fhl1 is up-regulated in cancers such as laryngeal carcinomas, gastric signet ring cell carcinoma, lung adenocarcinoma, gastric cancer, acute promyelocytic leukemia, and glioblastoma, among others. It is important to highlight that considering, the significant differences observed in in the expressions of Fhl1 between different tumors; Fhl1 is advised as a prognostic marker and could be a potential target for cancer therapy research including NHL.

  • Could you provide more details about the biological significance of the 106 proteins that were found to be differentially expressed in the lymph nodes of Balb/c mice? Specifically, how do these proteins relate to the therapeutic effects of IA and what part do they play in the pathophysiology of non-Hodgkin lymphoma?

ANSWER: All these information was include in discussion and conclusions sections

106 proteins showed significant expression differences (Table 8 and Table 9), including 66 down regulated proteins and 40 up regulated proteins. After all DEPs altered by IA treatment were subjects to PPI analysis, showing that “Four and a half LIM domains protein 1” (Fhl1) was the most important core protein (Figure 6), it showed directly or indirectly interaction with others proteins including, Il1rap, Ifi44, Timd4, Apoa4, and Fabp3 as well as Myh3, Eno 2, and H4c11. Myh3, Eno 2, and H4c11, were reported recently could be associated with the appearance of NHL. In tumors, the expression of Fhl1 is up regulated or down regulated and plays a role in promoting or inhibiting tumor development. The expression of Fhl1 is down regulated in several cancers such as lung, prostate, breast, ovarian, colon, thyroid, brain, kidney, liver, and melanoma, as well as oral cancers. In contrast, the expression of Fhl1 is up-regulated in cancers such as laryngeal carcinomas, gastric signet ring cell carcinoma, lung adenocarcinoma, gastric cancer, acute promyelocytic leukemia, and glioblastoma, among others.

Finnaly, the results of bioinformatic approaches suggest and provide additional evidence that support the therapeutic antitumor potential of IA. However, our research was limited by the lack of experimental validation particularly in the bioinformatics approaches. Also, in the translatability of mouse NHL models to human disease. Moreover, clinical studies focusing on Fhl1 as a key hub gene for early NHL diagnosis and as a potential target in precision medicine for NHL. Therefore, future studies are needed to investigate the role and impact of Fhl1 on patients with NHL and to elucidate its precise underlaying mechanisms. Also, systematic studies are necessary to validate the protein targets, changes at the protein level in treated lymph nodes and role of specific proteins in NHL. These studies including immunohistochemistry, Western blotting, ELISA, and miRNA expression among others.

  • What particular techniques were employed to evaluate IA's toxicity, and how do the results lend credence to its possible application as a therapeutic agent in NHL, particularly with regard to its pharmacokinetic characteristics and interactions with cancer-related proteins?

ANSWER:

To determine the physicochemical, pharmacokinetic and toxicological properties of IA and MTX, ADMETlab [68], SwissADME [69], admetSAR [70] and PROTOX [71] servers were used.

It is important to highlight that in recent years the bioinformatic analysis to predict the ADMET properties of compounds under drugs development has increased and has become an indispensable tool in the development of new drug candidates [13,18,19]. In relation to metabolism of IA and MTX, several CYP450 isoforms were evaluated to know if the molecules were substrate and/or inhibitor of them. Results suggests that both compounds can be metabolized by CYP3A4 and did not act like inhibitor of any CYP isoforms evaluated. In respect to distribution, the results suggests that IA must have a better volume of distribution than MTX and their excretion suggests that IA may have a higher half-life than MTX. Finally, results of toxicity suggests that MTX may be most toxic and cause hepatotoxicity, neurotoxicity and respiratory toxicity. In addition, the median lethal dose to humans predicted that MTX is most lethal than IA in agreement with category class I and IV, respectively. However, our research was limited by the lack of experimental validation particularly in the bioinformatics approaches. Moreover, clinical studies focusing on IA and its potential as antitumor agent for the treatment in NHL.

Introduction:

  • Mechanistic Understanding: What specific pathways underlie incomptine A's (IA) cytotoxic effects against various cancer cell types, and how does IA affect NF-kB expression and apoptosis in Non-Hodgkin lymphoma cells?

ANSWER:

In this work, NF-kB expresión was not found. However, in relation to cytotoxic properties IA exhibited dose-dependent activity against four subtypes of NHL cells (U-937, Farage, SU-DHL-2, and REC-1); In the case of U937 cells, regulates NF-kB expression, induces apoptosis, induces production of reactive oxygen species, as well as induces the differential protein expression of cytoskeleton proteins and glycolytic enzymes in U-937 cells and non-Hodgkin lymphoma in mice [10, 16]. In these sense, at doses of 5 mg/kg and 10 mg/kg IA cause DEPs such cytoskeleton proteins and   affect glycolytic procces in NHL model, 

  • In light of its documented cytotoxicity and low toxicity, how effective is IA in comparison to more conventional therapies for Non-Hodgkin lymphoma, such as methotrexate, in terms of therapeutic results and adverse effect profiles?

ANSWER

Results suggests that both compounds can be metabolized by CYP3A4 and did not act like inhibitor of any CYP isoforms evaluated. In respect to distribution, the results suggests that IA must have a better volume of distribution than MTX and their excretion suggests that IA may have a higher half-life than MTX. Finally, results of toxicity suggests that MTX may be most toxic and cause hepatotoxicity, neurotoxicity and respiratory toxicity. In addition, the median lethal dose to humans predicted that MTX is most lethal than IA in agreement with category class I and IV, respectively.

It is important to highlight that the result of toxicity obtained in this study to IA, was in agreement with the in vivo toxicity reported previously and confirms the predicted findings by bioinformatic [32]. Moreover, clinical studies Phase I and II are necceary focusing on IA and its potential as antitumor agent for the treatment in NHL.

  • Proteomic and Pharmacological Insights: How might the proteomic and network pharmacology methods used in this study help guide future NHL research and treatment plans? What new biomarkers and therapeutic targets were discovered?

ANSWER:

Moreover, clinical studies Phase I and II to obtain samples of blood and performed proteomic and network pharmacology and see the translatability of mouse NHL models to human disease, including therapeutic targets and biomarkers. this is first report que involves to Il1rap, Apoa4, Fabp3, IFi44, Timd4, and Fhl1 with of NHL

Materials:

  • Please write materials as Company Name (City, Country), especially for chemical analysis assessment which used in the study.

ANSWER:

Chemicals and Instrumentation

Triethylammonium bicarbonate buffer (1.0 M, pH 8.5±0.1), tris (2-carboxyethyl) phosphine hydrochloride solution (0.5 M, pH 7.0), iodoacetamide (IAA), formic acid (FA), acetonitrile (ACN), methanol (Sigma), trypsin from bovine pancreas (Promega), ultrapure water (Millipore), TMT 6-plex Isobaric Label Reagent, and Pierce Quantitative Colorimetric Peptide Assay (Thermo Fisher Science) were used in this work. Ultimate 3000 nano UHPLC system coupled to a Q Exactive HF MS equipped with a Nano spray Flex Ion Source (Thermo Scientific, Waltham, MA). TMT-based Quantification Analytical Service from Creative Proteomics (NY, USA).

4.3. Cell culture conditions

U-937 cells line were acquired from the American Type Culture Collection (CRL-1593,2, histiocytic lymphoma). To develop the mice model U-937 cells were cultured at 37 °C in RPMI 1640 culture medium (GIBCO Cat: 11875-093) supplemented with 5% fetal bovine serum (GIBCO Cat: 16000044), streptomycin (100 μg/mL)/penicillin (100 U/mL), and 5% CO2. In vivo tests were realized using cell cultures at a density of 2.5 x 106 cells in T75 flasks (Invitrogen, Waltham, MA, USA). Cell viability was determined by the trypan blue exclusion test. Cells were resuspended in fresh medium 24 h before treatments to ensure the exponential growth.

Methodology:

  • Please cite and have template “https://doi.org/10.1111/jfpp.16028 “for statistical analysis which is lack in materials and methods.

ANSWER:

Additional text was included to Antilymphoma Test (lines 635 to 639), Cytotoxic Activity  (lines 607 to 610), pathways enrichment (lines 699 to 707) and differential expression (lines 676 to 679; 683 to 686).

Statistical analysis of the data was performed using one-way ANOVA, as well as Dunnett´s multiple comparison tests with a value of p < 0.05 to establish a significant difference between the study groups.

Dotplot and three onthologies plots enrichment was performed with the ClusterProfiler package of the R v4.2.2 tool, following the steps proposed by the developer, using the GO databases (BP, MF, CC) and FDR < 0.05 as the value of statistical significance. Additionally, enrichment networks were performed in Cytoscape using the ClueGo as well as STRING plugins (confidence cutoff of 0.4), for both analyses, following the developer's manual using KEGG and GO databases and setting a p < 0.05 as the statistical significance criterion.  In parallel, we performed a functional enrichment analysis, also known as over-representation analysis (ORA), using the gProfiler tool available online, for which we used the parameters preset by the program

Data analytical report were carried out by Analytical Service from Creative Proteomics (New York, NY, USA) with p < 0.05 (indicates > 95% confidence of change in protein concentration respect of the magnitude of change) selected to designate differentially expressed proteins.

 It is important to highlight that fold change cutoffs at protein level accepted with biological meaning are defined as 1.5 times up or down. Therefore, the screening criteria to the relative quantitation of proteins in this work

Results:

  • All Tables and Figures: The alphabetical statistical letters for the means should all be modified such that the greatest number has the letter a and as the numbers go lower, letters b, c
  • ANSWER:

All Tables and figures were modified in aggrement with reviewers 1 and 2.

  • Interpretation of Pathway Analysis: What role do the enriched pathways—such as muscle contraction and glycolysis—found in the Gene Ontology and KEGG analyses play in understanding how incomptine A works to treat non-Hodgkin lymphoma, and what does this mean for potential future treatment approaches?

ANSWER:

Our contribution suggest that IA may be a compound with potential therapeutic against NHL. Its antitumoral properties may be associated in processes such as muscle contraction, glycolysis, hemostasis, epigenetic regulation of gene expression, transport of small molecules, neutrophil extracellular trap formation, adrenergic signaling in cardiomyocytes, systemic lupus erythematosus, alcoholism, platelet activation, signaling and aggregation.

Additional assays including immunohistochemistry, Western blotting, ELISA, and miRNA expression among others are mandatory to validate the results obtained here. Also, others assays are required to explore more molecular mechanisms of as IA promotes lymphoma cell death in NHL such as apoptosis (i.e., caspase activity, TUNEL staining, and annexin V/PI flow cytometry) and effect on glycolytic pathway (i.e., expression of aldolase A, lactate dehydrogenase, and hexokinase II).            Given the aforementioned, we propose that IA can be considered as a promising molecule with an exceptional pharmacological profile. Its potential application in the future as a novel pharmacological treatment for NHL, or as a foundation for the development of new molecule derivates to enhance its efficacy in treating this disease, warrants continue with the investigation of this molecule to acquire more information from it.

  • Molecular Docking Results: Could incomptine A (IA) be a new treatment option for NHL? What are the implications of molecular docking studies that demonstrate IA has a higher binding affinity to differentially expressed proteins than methotrexate?

ANSWER:

A paragraph that include te information suggested was added in lines 516-521 and 523, as follows:

This technique has gained significant importance in the development of novel treatments for various diseases. This computational technique predicts the binding affinity of ligands to receptor proteins. The assay yields free Gibbs energy (ΔG) values, which indicate the exergonic nature of the reaction. A more negative ΔG value suggest a higher probability of lingand-protein union and interaction [61-62].

  • Functional Validation: How will the down-regulated genes found in the Reactome analysis be validated experimentally, and how will these results influence the therapeutic potential of incomptine A in clinical settings?

ANSWER: Additional text was included lines 543 to 553 and  781 to 789

Additional assays including immunohistochemistry, Western blotting, ELISA, and miRNA expression among others are mandatory to validate the results obtained here. Also, others assays are required to explore more molecular mechanisms of as IA promotes lymphoma cell death in NHL such as apoptosis (i.e., caspase activity, TUNEL staining, and annexin V/PI flow cytometry) and effect on glycolytic pathway (i.e., expression of aldolase A, lactate dehydrogenase, and hexokinase II).            Given the aforementioned, we propose that IA can be considered as a promising molecule with an exceptional pharmacological profile. Its potential application in the future as a novel pharmacological treatment for NHL, or as a foundation for the development of new molecule derivates to enhance its efficacy in treating this disease, warrants continue with the investigation of this molecule to acquire more information from it.

However, our research was limited by the lack of experimental validation particularly in the bioinformatics approaches. Also, in the translatability of mouse NHL models to human disease. Moreover, clinical studies focusing on Fhl1 as a key hub gene for early NHL diagnosis and as a potential target in precision medicine for NHL. Therefore, future studies are needed to investigate the role and impact of Fhl1 on patients with NHL and to elucidate its precise underlaying mechanisms. Also, systematic studies are necessary to validate the protein targets, changes at the protein level in treated lymph nodes and role of specific proteins in NHL. These studies including immunohistochemistry, Western blotting, ELISA, and miRNA expression among others.

Discussion:

  • Discussion text must grammar improve and in some cases it is very weak and maybe there is no discussion at all.

ANSWER:

Additional texts were include in Discussion section.

  • Given its lower toxicity and superior ADMET profiles, how exactly does incomptine A (IA) treat non-Hodgkin lymphoma more effectively than methotrexate (MTX)?

ANSWER:

The complete analysis of ADMET calculation suggest that IA may have better absorption when it is administered orally than MTX, which also is reflected in a better distribution of the drug, in respect to metabolism this parameter was similar than MTX, and the excretion calculated for IA suggest that it may have a longer T1/2 than MTX. These results may support the important activity described on this study being IA more active than MTX on NHL treatment. Moreover, the toxicity for both molecules were determined, in this sense, we observe that IA may have lower toxicity than MTX with a predicted human lethal dose 50 (LD50) of 3 mg/kg for IA in comparison to 1330 mg/kg for MTX this result suggest a wide therapeutic window for the use of IA in comparison to MTX, and classifies to IA in class IV (300 < LD50 ≤ 2000 mg/kg) in comparison to MTX which is collocated in class I, which is considered fatal by ingestion (LD50 ≤ 5 mg/kg). MTX is known as one of the most widely used anticancer agent [29-31]. It is important to highlight that the result of toxicity obtained in this study to IA, was in agreement with the in vivo toxicity reported previously and confirms the predicted findings by bioinformatic [32]. In contrast, it is also known that the use of MTX may generate side effects such as nephrotoxicity, muscle pain, red eyes, swollen gums and hair lost among others [33,34], also, has been reported that high doses of MTX may be toxic to humans and mice [5,35] This is according to some predictions obtained from ADMET studies, and represent an advantage in the use of IA as a drug for the treatment of cancer. Also, MTX commonly its route of administration is intravenous being painful for the patients [36], in comparison, IA represent one important alternative due to our results suggest better orally absorption with important pharmacological anticancer activity, and perhaps with lower side effects than MTX. Finally, the drug likeness analysis supports the information showed on ADMET study, due to it suggest that IA fulfills with the necessary criteria to be an important candidate for the development of an orally drug.

  • What more research is required to examine the potential of "Four and a half LIM domains protein 1" (Fhl1) as a therapeutic target and prognostic biomarker for NHL, and how might this impact future drug development strategies, considering the study's identification of its significant role?

ANSWER

Clinical studies focusing on Fhl1 as a key hub gene for early NHL diagnosis and as a potential target in precision medicine for NHL. Therefore, future studies are needed to investigate the role and impact of Fhl1 on patients with NHL and to elucidate its precise underlaying mechanisms. Also, systematic studies are necessary to validate the protein targets, changes at the protein level in treated lymph nodes and role of specific proteins in NHL. These studies including immunohistochemistry, Western blotting, ELISA, and miRNA expression among others.

  • How can we better understand the biological mechanisms underlying NHL by combining proteomic, bioinformatics, and molecular docking studies? What does this mean for the development of novel therapeutic agents like IA?

ANSWER:

We add a paragraph according to your suggestion in lines 549-553, as follows:

Given the aforementioned, we propose that IA can be considered as a promising molecule with an exceptional pharmacological profile. Its potential application in the future as a novel pharmacological treatment for NHL, or as a foundation for the development of new molecule derivates to enhance its efficacy in treating this disease, warrants continue with the investigation of this molecule to acquire more information from it.

Conclusions:

  • Conclusion is very general, try to make it more scientific, comprehensive and concise in detail, especially.

ANSWER:

Conclusion was improved

Additional text was included

However, our research was limited by the lack of experimental validation particularly in the bioinformatics approaches. Also, in the translatability of mouse NHL models to human disease. Moreover, clinical studies focusing on Fhl1 as a key hub gene for early NHL diagnosis and as a potential target in precision medicine for NHL. Therefore, future studies are needed to investigate the role and impact of Fhl1 on patients with NHL and to elucidate its precise underlaying mechanisms. Also, systematic studies are necessary to validate the protein targets, changes at the protein level in treated lymph nodes and role of specific proteins in NHL. These studies including immunohistochemistry, Western blotting, ELISA, and miRNA expression among others.

References: It is OK.

Also;

New version was reviwed by all authors and a colleague native speaker of english

Round 2

Reviewer 1 Report

Comments and Suggestions for Authors

The manuscript has significantly improved, and the authors have admitted their flaws and limitations. However, with no experimental validation of predicted protein targets or functional mechanisms, the manuscript cannot achieve the level of completion usually required for acceptance. However, suppose the journal is willing to accept this study at a lower level of experimental depth. In that case, it may be an option for publication with minor additional revisions based on the level of original work (in vivo work and bioinformatics data). If experimental validation will not be possible, as the author mentioned in the reply, I suggest the authors remove all mechanistic assertions in the title, abstract, and discussion and clarify that they are mainly making predictive/hypothesis-generating conclusions.

The second point is that the manuscript recognises a 39% similarity, well above the acceptable level. The manuscript requires significant revision to handle the plagiarism/self-plagiarism issue before we can consider the article again for peer review.

Author Response

August 18, 2025

To the Editorial Board of Pharmaceuticals (MDPI)

Dear Editor/Reviewer:

Regard to the manuscript:

Reviewer 1,

Thank you for your valuable time and effort in reviewing this manuscript. Your suggestions have significantly enhanced the quality of our article. We have meticulously addressed your comments as detailed below for the manuscript “Pharmaceuticals-3815701”.

Following the revisions, we would appreciate your consent to publish the manuscript after the minor revisions mentioned above.

  1. If experimental validation will not be possible, as the author mentioned in the reply, I suggest the authors remove all mechanistic assertions in the tittle, abstract and discussion and clarify that they are mainly making predictive/hypotesis-generating conclusions.

Answer. Your suggestions have been instrumental in strengthening the machanistic claims of the work We have redirected the research towards primarly predicitive or hypothesis generating conclusions as follows:

  • In line 545 the paragraph: as well as know that process and mechanism is involved in NHL and other cancers. Have been deleted.
  • In line 547, the paragraph : Also, other assays are required to explore more molecular mechanisms of as IA promotes lymphoma cell death in NHL such as apoptosis. Have been change by: Also, others assays are required to explore the way as IA promotes lymphoma cell death in NHL such as apoptosis
  • In line 779-782, the paragraph: Also, our findings could have potential clinical value in the diagnosis and therapeutic treatment of NHL and provide valuable information new studies of the molecular mechanism of NHL. Have been change by : Additionally, our findings could have potential clinical value in the diagnosis and therapeutic treatment of NHL, beyond enriching the existing information of the pharmacological activity of IA on NHL.
  • In line 786-788: the paragraph: Therefore, future studies are needed to investigate the role and impact of Fhl1 on patients with NHL and to elucidate its precise underlaying mechanisms. Have been changed by: Therefore, future studies are needed to investigate the role and impact of Fhl1 on patients with NHL and to elucidate its precise underlaying their effects.
  1. The second point is that the manuscript recognizes a 39% similarity, well above the acceptable level. The manuscript requires significant revision to handle the plagiarism/self-plagiarism issue before we can consider the article again for peer review.

Regarding the second point, the manuscript recognizes a similarity of 39%, well above the acceptable level. We analyzed the manuscript using the iThenticate Turnitin tool, and after a thorough review to address the similarity issue, we found that the percentage is increased, for example in tables, in terms that cannot be replaced even if they have been used in other publications (e.g. Hepatotoxicity, Mutagenicity, Inactive, Caco-2 permeability, CYP2C9, substrate, Molecular weight, Number of rotating links, etc.), protein names (e.g. DNA replication licensing factor MCM4, PDZ and LIM domain protein 5, High mobility group nucleosome-binding domain containing protein 5, Histone H2B type 1-P, Myosin light chain 3, Ig kappa chain V-V region HP 93G7, etc.), figure captions that describe the databases or tools used (e.g. GO, REAC, KEGG database, gene ontology cellular component, Kyoto Encyclopedia of Genes, etc.), main text, adscriptions or recurring words (e.g., Non-Hodgkin lymphoma, lymphatic nodes, lymphocytes,  incomptine A,  male Balb/c mice, U-937, Farage, SU-DHL-2, and REC-1cells, cancer, organs, malignant transformation, proliferation, apoptosis, heliangolide sesquiterpene lactone, Decachaeta incompta, 4T1, MDA-MB-231, SK-BR-3, T-47D, MCF7, MCF10A, reactive oxygen species, differential protein expression, cytoskeleton proteins, glycolytic enzymes); and there is mainly similarity in the section of Material and Methods (Isolation of Incomptine, Chemicals and Instrumentation, Cell culture conditions, Cytotoxic Activity, Animals, Antilymphoma Test, Non-Hodgkin’s Lymphoma Protein Expression Induced Through IA or MTX, Nano UHPLC-MS/MS Analyses, Protein Identification, Differentially Expressed Protein Analysis, Bioinformatic Methodology, Comparison of Shared Processes and Molecules, In Silico Physicochemical, Pharmacokinetic and Toxicological Properties, Molecular Docking Studies), which does not have much room for paraphrasing or replacing, since that is what was done, and we are sure that the similarity is due to our previous articles part I and part II of this series of findings on the potential of Incomptina A, in the same Pharmaceuticals Journal. Filtering headings, titles, page numbers, methodology, references, affiliations, some sections (Data Availability, Acknowledgments, Author Contributions, Funding, Institutional Review Board Statement, Informed Consent Statement, etc., the percentage is less than 14%, which is acceptable for any plagiarism-free journal (We attach the iThenticate Turnitin report at the end of the cover letter).

This article is an original work, and no portion of results, conclusion or discussion of this study has been published or submitted for publication elsewhere. The manuscript coincides to the Special Issue "Biomedical Properties, Developments and Therapeutic Potential of Sesquiterpenoid Lactones and Natural Compounds", special issue in Pharmaceuticals", we hereby confirm that this research fits the aims and topics of your journal.

We uploaded the final proofread version of our manuscript attending all the changes in the document.

All authors agree to the final version. We Carefully check the names and affiliations to be correct.

Thank you very much for your consideration and patience.

Sincerely yours,

Dr. Prof. Normand García Hernández,

Corresponding author.
